# Mastering Multiple-Expert Routing: Realizable $\mathcal{H}$-Consistency and Strong Guarantees for Learning to Defer

Anqi Mao [1]   Mehryar Mohri [2 1]   Yutao Zhong [2]

## Abstract

The problem of learning to defer with multiple experts consists of optimally assigning input instances to experts, balancing the trade-off between their accuracy and computational cost. This is a critical challenge in natural language generation, but also in other fields such as image processing, and medical diagnostics. Recent studies have proposed surrogate loss functions to optimize deferral, but challenges remain in ensuring their consistency properties. This paper introduces novel surrogate loss functions and efficient algorithms with strong theoretical learning guarantees. We address open questions regarding realizable $\mathcal{H}$-consistency, $\mathcal{H}$-consistency bounds, and Bayes-consistency for both single-stage (jointly learning predictor and deferral function) and two-stage (learning only the deferral function with a fixed expert) learning scenarios. For single-stage deferral, we introduce a family of new realizable $\mathcal{H}$-consistent surrogate losses and further prove $\mathcal{H}$-consistency for a selected member. For two-stage deferral, we derive new surrogate losses that achieve realizable $\mathcal{H}$-consistency, $\mathcal{H}$-consistency bounds, and Bayes-consistency for the two-expert scenario and, under natural assumptions, multiple-expert scenario. Additionally, we provide enhanced theoretical guarantees under low-noise assumptions for both scenarios. Finally, we report the results of experiments using our proposed surrogate losses, comparing their performance against existing baselines.

## 1. Introduction

The performance of learning algorithms can be substantially improved by redirecting uncertain predictions to domain specialists or advanced pre-trained systems. These specialists may include individuals with deep expertise in specific areas or highly capable, though computationally intensive, pre-trained models. Selecting the appropriate expert requires careful consideration of both accuracy and computational cost, which may vary depending on the instance or class label in question.

How can input instances be optimally assigned to the most suitable expert from a diverse set, balancing these considerations? Can allocation strategies be learned from past experience? These questions form the core challenge of *learning to defer with multiple experts*, a problem that comes up across various fields. In natural language generation, particularly with large language models (LLMs), this challenge has been highlighted as a critical issue (Wei et al., 2022; Bubeck et al., 2023), essential both for reducing hallucinations and improving efficiency. However, the problem extends to other domains, including image annotation, medical diagnostics, economic forecasting, and computer vision, among others.

The problem of learning to defer accurately has been investigated in a number of recent studies (Mozannar & Sontag, 2020; Verma & Nalisnick, 2022; Charusaie et al., 2022; Mozannar et al., 2023; Mao et al., 2024a;i). It is typically formulated using a deferral loss function that incorporates instance-specific costs associated with each expert. Directly optimizing this loss function is intractable for the hypothesis sets commonly used in applications. Thus, learning-to-defer algorithms rely on optimizing a surrogate loss function instead, that serves as a proxy for the original target loss function. Yet, what guarantees can we rely on when optimizing such surrogate loss functions?

This question, which involves analyzing the consistency guarantees of surrogate losses with respect to the deferral loss, has been studied under two main scenarios (Mao, 2025): the *single-stage* scenario, where a predictor and a deferral function are jointly learned (Mozannar & Sontag, 2020; Verma & Nalisnick, 2022; Charusaie et al., 2022;

---
[1]Courant Institute of Mathematical Sciences, New York, NY; [2]Google Research, New York, NY. Correspondence to: Anqi Mao <aqmao@cims.nyu.edu>, Mehryar Mohri <mohri@google.com>, Yutao Zhong <yutaozhong@google.com>.

*Proceedings of the 42$^{nd}$ International Conference on Machine Learning*, Vancouver, Canada. PMLR 267, 2025. Copyright 2025 by the author(s).

Mozannar et al., 2023; Mao et al., 2024a), and a *two-stage* scenario, where the predictor is pre-trained and fixed as an expert, and while only the deferral function is subsequently learned (Mao et al., 2023a).

In particular, Mozannar & Sontag (2020), Verma & Nalisnick (2022), and Charusaie et al. (2022) proposed surrogate loss functions for the single-stage *single-expert* case by generalizing the cross-entropy loss, the one-versus-all loss, and, more generally, a broad family of surrogate losses for multi-class classification in the context of learning to defer. However, Mozannar et al. (2023) later showed that these surrogate loss functions do not satisfy *realizable $\mathcal{H}$-consistency*. They suggested an alternative surrogate loss that achieves this property but left open the question of whether it was also Bayes-consistent. This was later resolved by Mao et al. (2024i), who introduced a broader family of surrogate losses that simultaneously achieve Bayes-consistency, realizable $\mathcal{H}$-consistency, and $\mathcal{H}$-*consistency bounds*.

For the single-stage multiple-expert setting, Verma et al. (2023) were the first to extend the surrogate loss proposed in (Verma & Nalisnick, 2022) and (Mozannar & Sontag, 2020) to accommodate multiple experts. Building on this, Mao et al. (2024a) further generalized the surrogate loss from (Mozannar & Sontag, 2020), introducing a broader family of surrogate losses tailored to the multiple-expert case. Furthermore, Mao et al. (2024a) proved that their surrogate losses benefit from $\mathcal{H}$-consistency bounds in the multiple-expert case, thereby ensuring Bayes-consistency. However, these loss functions are not realizable $\mathcal{H}$-consistent even in the single-expert case, as they are extensions of the earlier loss functions. Can surrogate loss functions for single-stage multiple-expert deferral be derived that admit realizable $\mathcal{H}$-consistency, along with $\mathcal{H}$-consistency bounds and Bayes-consistency?

In the *two-stage* scenario, Mao et al. (2023a) introduced surrogate losses that are Bayes-consistent and realizable $\mathcal{H}$-consistent for constant costs. However, their realizable $\mathcal{H}$-consistency does not extend to cost functions of interest, which are based on classification error. Can we derive surrogate loss functions for two-stage multiple-expert deferral that achieve realizable $\mathcal{H}$-consistency $\mathcal{H}$-consistency bounds, and Bayes-consistency for cost functions based on classification error?

It is also important to analyze properties beyond realizable $\mathcal{H}$-consistency whose guarantees hold under deterministic assumptions, while $\mathcal{H}$-consistency bounds extend to arbitrary distributions. Can we offer guarantees for intermediate cases, particularly for distributions satisfying low-noise assumptions?

This work addresses all of these questions. Note that these challenges are more significant and more complex in the

multiple-expert case than the single-expert case. In Section 3, we derive realizable surrogate losses for single-stage multiple-expert deferral. We begin by deriving an alternative formulation for the deferral loss, which serves as the foundation for defining a novel family of surrogate losses (Section 3.1). We then establish the realizable $\mathcal{H}$-consistency of these loss functions under a mild assumption (Section 3.2). Finally, we prove $\mathcal{H}$-consistency bounds for a specific surrogate loss within this family (Section 3.3).

In Section 4, we define realizable surrogate losses for two-stage multiple-expert deferral. We first prove that a family of surrogate losses is realizable $\mathcal{R}$-consistent, $\mathcal{R}$-consistent, and Bayes-consistent in the two-expert case (Section 4.1). Next, we extend this family of surrogate losses and their consistency guarantees to the multiple-expert case under natural assumptions (Section 4.2).

In Section 5, we present enhanced bounds for the intermediate case of distributions satisfying low-noise assumptions. We provide guarantees for both the single-stage (Section 5.1) and two-stage (Section 5.2) scenarios.

Finally, in Section 6, we report the results of experiments using the proposed surrogate losses, comparing their performance against existing baselines.

For a summary of previous work on consistent multiple-expert deferral and a more detailed discussion of related work, see Table 1 and Appendix A, respectively. We begin with a formal introduction to the learning problems and key concepts.

## 2. Preliminaries

**Single-stage multiple-expert deferral.** Let $\mathcal{X}$ denote an input space, and let $\mathcal{Y} = [n] \coloneqq \{1, \ldots, n\}$ represent a label space with $n \geq 2$ labels, as in the standard multi-class classification setting. In the *single-stage multiple-expert deferral* scenario, the learner has the option to predict the true label or defer the prediction to one of $n_e$ pre-defined experts. In this scenario, the label space $\mathcal{Y}$ is augmented with $n_e$ additional labels, $\{n + 1, \ldots, n + n_e\}$, corresponding to the $n_e$ experts $g_1, \ldots, g_{n_e}$. Each expert may represent a human expert or a pre-trained model. Specifically, each expert can be expressed as a function mapping $\mathcal{X} \times \mathcal{Y}$ to $\mathbb{R}$. Let $\overline{\mathcal{Y}} = [n + n_e]$ denote the augmented label set, and let $\mathcal{H}$ be a hypothesis set of functions mapping $\mathcal{X} \times \overline{\mathcal{Y}}$ to $\mathbb{R}$. Let $\mathcal{H}_{\mathrm{all}}$ represent the family of all such measurable functions. The goal of the learner is to select a hypothesis $h \in \mathcal{H}$ that minimizes the following *single-stage deferral loss function*, $\mathsf{L}_{\mathrm{def}}$, defined for any $h \in \mathcal{H}_{\mathrm{all}}$ and $(x, y) \in \mathcal{X} \times \mathcal{Y}$ as follows:

$$\mathsf{L}_{\mathrm{def}}(h, x, y) = \mathbb{1}_{\mathsf{h}(x) \neq y} \mathbb{1}_{\mathsf{h}(x) \in [n]} + \sum_{j=1}^{n_e} c_j(x, y) \mathbb{1}_{\mathsf{h}(x) = n + j},$$

where $\mathsf{h}(x) = \operatorname{argmax}_{y \in [n + n_e]} h(x, y)$ is the prediction associated with $h \in \mathcal{H}$ for an input $x \in \mathcal{X}$, using an arbitrary

*Table 1.* Summary of Previous Work on Consistent Multiple-Expert Deferral.

| Related Work | Deferral Setting | Realizable $\mathcal{H}$-Consistency | Bayes-Consistency | $\mathcal{H}$-Consistency Bounds |
|---|---|---|---|---|
| (Verma et al., 2023) | Single-stage | No | Yes | Yes |
| (Mao et al., 2024a) | Single-stage | No | Yes | Yes |
| (Mao et al., 2023a) | Two-stage | No | Yes | Yes |

but fixed deterministic strategy for breaking ties. When the prediction $\mathsf{h}(x)$ is in $\mathcal{Y}$, the incurred loss coincides with the multi-class zero-one classification loss. When the prediction $\mathsf{h}(x)$ equals $n + j$, the incurred loss is the cost of deferring to expert $g_j$, denoted by $c_j(x, y)$. The choice of this cost is highly flexible. A common choice is to define it as $g_j$'s classification error (Verma et al., 2023): $c_j(x, y) = \mathbb{1}_{\mathsf{g}_j(x) \neq y}$, where $\mathsf{g}_j(x) = \operatorname{argmax}_{y \in [n]} g_j(x, y)$ represents the prediction made by expert $g_j$ for input $x$.

**Two-stage multiple-expert deferral.** In the *two-stage multiple-expert deferral* scenario, a label predictor is assumed to have been learned during the first stage. The second stage involves a set of $n_e \geq 2$ predefined experts, denoted $g_1, \ldots, g_{n_e}$, which includes the first-stage label predictor. We define $\mathcal{Y}$ as $[n_e] \coloneqq 1, \ldots, n_e$. In this setup, the learner's objective is to select a suitable expert $g_j$ for each instance, considering both the expert's accuracy and inference cost. More formally, let $\mathcal{R}$ represent a hypothesis set of functions mapping $\mathcal{X} \times [n_e]$ to $\mathbb{R}$, and let $\mathcal{R}_{\mathrm{all}}$ denote the family of all such measurable functions. The goal is to choose a predictor $r \in \mathcal{R}$ that minimizes the following *two-stage deferral loss function*, $\mathsf{L}_{\mathrm{tdef}}$, defined for any $r \in \mathcal{R}_{\mathrm{all}}$ and $(x, y) \in \mathcal{X} \times \mathcal{Y}$ as follows:

$$\mathsf{L}_{\mathrm{tdef}}(r, x, y) = \sum_{j=1}^{n_e} c_j(x, y) \mathbb{1}_{\mathsf{r}(x) = j},$$

where $\mathsf{r}(x) = \operatorname{argmax}_{y \in [n_e]} r(x, y)$ represents the prediction associated with $r \in \mathcal{R}$ for an input $x \in \mathcal{X}$, using an arbitrary but fixed deterministic strategy for breaking ties. The cost $c_j(x, y)$ is incurred when the learner chooses to defer to expert $g_j$. As in the single-stage scenario, the choice of the cost is flexible and can be defined as $g_j$'s classification error (Mao et al., 2023a): $c_j(x, y) = \mathbb{1}_{\mathsf{g}_j(x) \neq y}$.

**Learning with surrogate losses.** Minimizing the single-stage or two-stage deferral losses is computationally intractable for most hypothesis sets due to their non-continuity and non-differentiability, as for the multi-class zero-one loss. Instead, we will consider surrogate losses for the deferral loss. Let $\mathcal{D}$ be a distribution over $\mathcal{X} \times \mathcal{Y}$ and let $\mathsf{L}$ be a loss function. The generalization error of a hypothesis $h \in \mathcal{H}$ is defined by $\mathcal{E}_{\mathsf{L}}(h) = \mathbb{E}_{(x,y) \sim \mathcal{D}}[\mathsf{L}(h, x, y)]$ and the best-in-class generalization error by $\mathcal{E}_{\mathsf{L}}^*(\mathcal{H}) = \inf_{h \in \mathcal{H}} \mathcal{E}_{\mathsf{L}}(h)$. We refer to the difference $\mathcal{E}_{\mathsf{L}}(h) - \mathcal{E}_{\mathsf{L}}^*(\mathcal{H})$ as the estimation error. When $\mathcal{H} = \mathcal{H}_{\mathrm{all}}$, we refer to $\mathcal{E}_{\mathsf{L}}^*(\mathcal{H}_{\mathrm{all}})$ as the Bayes error and $\mathcal{E}_{\mathsf{L}}(h) - \mathcal{E}_{\mathsf{L}}^*(\mathcal{H}_{\mathrm{all}})$ as the excess error. In the two-

stage case, these definitions extend naturally by replacing the hypothesis $h$ with $r$ and the hypothesis set $\mathcal{H}$ with $\mathcal{R}$. For clarity, we will introduce these concepts in the single-stage setting as an example, though they also apply to the two-stage case.

A necessary and fundamental property of a surrogate loss is *Bayes-consistency* (Zhang, 2004a; Bartlett et al., 2006; Zhang, 2004b; Tewari & Bartlett, 2007; Steinwart, 2007; Mozannar & Sontag, 2020; Verma & Nalisnick, 2022), which means that minimizing the excess error of the surrogate loss $\mathsf{L}$ leads to minimizing the excess error of the deferral loss $\mathsf{L}_{\mathrm{def}}$.

**Definition 2.1.** A surrogate loss $L$ is *Bayes-consistent with respect to* $\mathsf{L}_{\mathrm{def}}$ if for any sequence of predictors $\{(h_n)\}_{n \in \mathbb{N}} \subset \mathcal{H}_{\mathrm{all}}$ such that $[\mathcal{E}_{\mathsf{L}}(h_n) - \mathcal{E}_{\mathsf{L}}(\mathcal{H}_{\mathrm{all}})]$ converges to zero as $n \to +\infty$, $[\mathcal{E}_{\mathsf{L}_{\mathrm{def}}}(h_n) - \mathcal{E}_{\mathsf{L}_{\mathrm{def}}}(\mathcal{H}_{\mathrm{all}})]$ converges to zero too.

Bayes-consistency does not take into account the specific hypothesis set $\mathcal{H}$ adopted in practice and assumes that optimization is performed over the family of all measurable functions. Consequently, a hypothesis-dependent guarantee, such as *realizable $\mathcal{H}$-consistency* (Long & Servedio, 2013; Zhang & Agarwal, 2020; Mozannar et al., 2023; Mao et al., 2024i) and $\mathcal{H}$-*consistency bound* (Awasthi et al., 2022a;b; Mao et al., 2023f) (see also (Awasthi et al., 2021a;b; 2023; 2024; Mao et al., 2023b;e; Zheng et al., 2023; Mao et al., 2023c;d; 2024f;g; Cortes et al., 2024; 2025; Mao et al., 2025; Zhong, 2025)), is more informative and relevant.

**Definition 2.2.** We will say that a surrogate loss $\mathsf{L}$ is *realizable $\mathcal{H}$-consistent with respect to* $\mathsf{L}_{\mathrm{def}}$ if for any *realizable distribution* (a distribution for which there exists $h^* \in \mathcal{H}$ satisfying $\mathcal{E}_{\mathsf{L}_{\mathrm{def}}}(h^*) = 0$), if $\hat{h}$ is in $\operatorname{argmin}_{h \in \mathcal{H}} \mathcal{E}_{\mathsf{L}}(h)$ then $\mathcal{E}_{\mathsf{L}_{\mathrm{def}}}(\hat{h}) = 0$.

Realizable $\mathcal{H}$-consistency is an asymptotic guarantee and is independent of Bayes-consistency. An $\mathcal{H}$-*consistency bound*, on the other hand, is non-asymptotic and always implies Bayes-consistency.

**Definition 2.3.** A surrogate loss $\mathsf{L}$ is said to admit an $\mathcal{H}$-*consistency bound*, if there exists a non-decreasing concave function $\Gamma \colon \mathbb{R}_+ \to \mathbb{R}_+$ with $\Gamma(0) = 0$, such that the following inequality holds for all $h \in \mathcal{H}$ and all distributions:

$$\mathcal{E}_{\mathsf{L}_{\mathrm{def}}}(h) - \mathcal{E}_{\mathsf{L}_{\mathrm{def}}}^*(\mathcal{H}) + \mathcal{M}_{\mathsf{L}_{\mathrm{def}}}(\mathcal{H})$$
$$\leq \Gamma(\mathcal{E}_{\mathsf{L}}(h) - \mathcal{E}_{\mathsf{L}}^*(\mathcal{H}) + \mathcal{M}_{\mathsf{L}}(\mathcal{H})),$$

where the *minimizability gap* $\mathcal{M}_{\mathsf{L}}(\mathcal{H})$ is defined as $\mathcal{M}_{\mathsf{L}}(\mathcal{H}) = \mathcal{E}_{\mathsf{L}}^*(\mathcal{H}) - \mathbb{E}_x[\inf_{h \in \mathcal{H}} \mathbb{E}_{y|x}[\mathsf{L}(h,x,y)]]$.

For convenience, we refer to $\mathcal{C}_{\mathsf{L}}(h,x) = \mathbb{E}_{y|x}[\mathsf{L}(h,x,y)]$ and $\mathcal{C}_{\mathsf{L}}^*(\mathcal{H},x) = \inf_{h \in \mathcal{H}} \mathcal{C}_{\mathsf{L}}(h,x)$ as the conditional error and best-in-class conditional error, respectively. We refer to the difference, $\mathcal{C}_{\mathsf{L}}(h,x) - \inf_{h \in \mathcal{H}} \mathcal{C}_{\mathsf{L}}(h,x)$, as the conditional regret and denote it by $\Delta\mathcal{C}_{\mathsf{L},\mathcal{H}}(h,x)$. The minimizability gap is the difference between the best-in-class generalization error and the expected best-in-class conditional error. It can be upper-bounded by the approximation error and vanishes when $\mathcal{H} = \mathcal{H}_{\mathrm{all}}$ (Awasthi et al., 2022a;b; Mao et al., 2024d). Thus, an $\mathcal{H}$-consistency bound implies Bayes-consistency. However, in the deferral setting, the minimizability gap may not vanish for a realizable distribution. Therefore, $\mathcal{H}$-consistency bounds do not imply realizable $\mathcal{H}$-consistency (Mao et al., 2024i).

# 3. Single-Stage Multiple-Expert Deferral

In this section, we derive realizable surrogate losses for single-stage multiple-expert deferral. We begin by proving an alternative formulation for $\mathsf{L}_{\mathrm{def}}$, which serves as the foundation for defining a novel family of surrogate losses. Next, we establish the realizable $\mathcal{H}$-consistency of these loss functions under a mild assumption. Finally, we establish $\mathcal{H}$-consistency bounds for a specific surrogate loss within this family.

## 3.1. New Surrogate Losses

We first prove that the following alternative expression holds for $\mathsf{L}_{\mathrm{def}}$. The proof is included in Appendix B.1.

**Lemma 3.1.** *The deferral loss can be expressed as follows:* $\forall(h,x,y) \in \mathcal{H}_{\mathrm{all}} \times \mathcal{X} \times \mathcal{Y}$,

$$\mathsf{L}_{\mathrm{def}}(h,x,y) = \left[\sum_{j=1}^{n_e} c_j(x,y) + 1 - n_e\right]1_{\mathsf{h}(x) \neq y} \quad (1)$$
$$+ \sum_{j=1}^{n_e}\left[1 - c_j(x,y)\right]1_{\mathsf{h}(x) \neq y}1_{\mathsf{h}(x) \neq n+j}.$$

We use this formulation to derive a new family of surrogate losses by replacing the indicator functions in (1) with smooth loss functions. The first indicator function, $1_{\mathsf{h}(x) \neq y}$, corresponds to the multi-class zero-one loss. Thus, a natural approach is to substitute it with a surrogate loss used in standard multi-class classification. In particular, we focus on the broad class of comp-sum losses (Mao et al., 2023f), which includes many well-known loss functions, such as the logistic loss. Comp-sum losses are defined for all $(h,x,y) \in \mathcal{H} \times \mathcal{X} \times \mathcal{Y}$ as:

$$\Psi\left(\frac{e^{h(x,y)}}{\sum_{y' \in \overline{\mathcal{Y}}} e^{h(x,y')}}\right), \quad (2)$$

where $\Psi\colon[0,1] \to \mathbb{R}_+ \cup \{+\infty\}$ is a non-increasing function. A specific choice of $\Psi$ is the family $\Psi_q$ defined for all $u \in (0,1]$ by $\Psi_q(u) = \frac{1}{q}(1 - u^q)$ for $q > 0$, and $\Psi_q(u) = -\log(u)$ for $q = 0$. For $q = 0$, $\Psi_q$ corresponds to the *(multinomial) logistic loss* (Verhulst, 1838; 1845; Berkson, 1944; 1951), while $q \in (0,1)$ corresponds to the *generalized cross-entropy loss* (Zhang & Sabuncu, 2018), and $q = 1$ to the *mean absolute error loss* (Ghosh et al., 2017). For the indicator function $1_{\mathsf{h}(x) \neq y}1_{\mathsf{h}(x) \neq n+j}$, we adopt the following modified comp-sum surrogate loss, as in (Mao et al., 2024i): $\Psi\left(\frac{e^{h(x,y)} + e^{h(x,n+j)}}{\sum_{y' \in \overline{\mathcal{Y}}} e^{h(x,y')}}\right)$. This leads to the following new family of surrogate losses for the deferral loss:

$$\mathsf{L}_{\Psi}(h,x,y) = \left[\sum_{j=1}^{n_e} c_j(x,y) + 1 - n_e\right]\Psi\left(\frac{e^{h(x,y)}}{\sum_{y' \in \overline{\mathcal{Y}}} e^{h(x,y')}}\right)$$
$$+ \sum_{j=1}^{n_e}\left[1 - c_j(x,y)\right]\Psi\left(\frac{e^{h(x,y)} + e^{h(x,n+j)}}{\sum_{y' \in \overline{\mathcal{Y}}} e^{h(x,y')}}\right), \quad (3)$$

where $\Psi\colon[0,1] \to \mathbb{R}_+ \cup \{+\infty\}$ is a decreasing function.

## 3.2. Realizable $\mathcal{H}$-Consistency

A hypothesis set $\mathcal{H}$ is said to be *closed under scaling* if, $(h \in \mathcal{H})$ implies $(\alpha h \in \mathcal{H})$ for any $\alpha \in \mathbb{R}$. This property is satisfied by many broad function classes, including linear functions and various families of neural networks. The following result shows that, under mild assumptions, our proposed surrogate losses are realizable $\mathcal{H}$-consistent when $\mathcal{H}$ is closed under scaling.

**Theorem 3.2.** *Assume $n_e \geq 2$ and $\mathcal{H}$ closed under scaling. Then, if $\Psi$ satisfies $\lim_{u \to 1^-} \Psi(u) = 0$ and $\lim_{u \to 0^+} \Psi(u) = 1$, then the surrogate loss $\mathsf{L}_{\Psi}$ is realizable $\mathcal{H}$-consistent with respect to $\mathsf{L}_{\mathrm{def}}$.*

The proof is given in Appendix B.2. Note that for $n_e = 1$, our surrogate loss formulation (3) coincides with that of $\mathsf{L}_{\mathrm{RL2D}}$ in the single-expert case by Mao et al. (2024i). In this special case, the following result holds.

**Theorem 3.3** (Theorem 4.1 in (Mao et al., 2024i)). *Assume $n_e = 1$ and $\mathcal{H}$ closed under scaling. Then, if $\Psi$ satisfies $\lim_{u \to 1^-} \Psi(u) = 0$, then, the surrogate loss $\mathsf{L}_{\Psi}$ is realizable $\mathcal{H}$-consistent with respect to $\mathsf{L}_{\mathrm{def}}$.*

Compared to the single-expert case where $n_e = 1$, an additional condition, $\lim_{u \to 0^+} \Psi(u) = 1$, is required for $\mathsf{L}_{\Psi}$ to be realizable $\mathcal{H}$-consistent with respect to $\mathsf{L}_{\mathrm{def}}$, as stated in Theorem 3.2. This condition rules out the case where $\Psi(u) = -\log(u)$ but is satisfied by functions $\Psi = \Psi_q$ when $q > 0$. We will verify in a simulated example in Section 6 that the surrogate loss $\mathsf{L}_{\Psi}$ with $\Psi = \Psi_q$ ($q > 0$) is realizable $\mathcal{H}$-consistent.

### 3.3. $\mathcal{H}$-Consistency Bounds

We now show that in the special case of $\Psi = \Psi_1$, that is $\Psi(u) = 1 - u$, $L_\Psi$ admits an $\mathcal{H}$-consistency bound and is Bayes-consistent with respect to $L_{\text{def}}$. For this case, the conditions $\lim_{u\to 1^-} \Psi(u) = 0$ and $\lim_{u\to 0^+} \Psi(u) = 1$ also hold. We denote the resulting surrogate loss as $L_{\text{mae}}$. It is defined as follows for any $(h, x, y) \in \mathcal{H}_{\text{all}} \times \mathcal{X} \times \mathcal{Y}$:

$$
\begin{aligned}
L_{\text{mae}}&(h, x, y) \\
&= \left[ \sum_{j=1}^{n_e} c_j(x, y) + 1 - n_e \right]\left( 1 - \frac{e^{h(x,y)}}{\sum_{y' \in \overline{\mathcal{Y}}} e^{h(x,y')}} \right) \\
&\quad + \sum_{j=1}^{n_e} \left[ 1 - c_j(x, y) \right]\left( 1 - \frac{e^{h(x,y)} + e^{h(x,n+j)}}{\sum_{y' \in \overline{\mathcal{Y}}} e^{h(x,y')}} \right). \quad (4)
\end{aligned}
$$

We say that a hypothesis set is *symmetric* if there exists a family $\mathcal{F}$ of functions mapping from $\mathcal{X}$ to $\mathbb{R}$ such that $\{[h(x, 1), \ldots, h(x, n + n_e)] : h \in \mathcal{H}\} = \mathcal{F}^{n+n_e}$. We say that a hypothesis set $\mathcal{H}$ is *complete* if for any $(x, y) \in \mathcal{X} \times \overline{\mathcal{Y}}$, the set of scores generated by $\mathcal{H}$ spans all real numbers: $\{h(x, y) \mid h \in \mathcal{H}\} = \mathbb{R}$. Commonly used hypothesis sets such as families of neural networks, linear functions, and all measurable functions, are both symmetric and complete.

**Theorem 3.4.** *Assume that $\mathcal{H}$ is symmetric and complete. Then, for all $h \in \mathcal{H}$ and any distribution, the following $\mathcal{H}$-consistency bound holds:*

$$
\begin{aligned}
\mathcal{E}_{L_{\text{def}}}&(h) - \mathcal{E}_{L_{\text{def}}}(\mathcal{H}) + \mathcal{M}_{L_{\text{def}}}(\mathcal{H}) \\
&\leq (n + n_e)\big(\mathcal{E}_{L_{\text{mae}}}(h) - \mathcal{E}_{L_{\text{mae}}}(\mathcal{H}) + \mathcal{M}_{L_{\text{mae}}}(\mathcal{H})\big).
\end{aligned}
$$

The proof is given in Appendix B.3.2. Theorem 3.4 includes the single-expert case (Theorem 4.3 in (Mao et al., 2024i)) as a special instance when $n_e = 1$. In the proof, we first characterize the conditional regret of the deferral loss, which is determined by the key quantities $p_{n+j} = \sum_{y \in \mathcal{Y}} p(y|x)(1 - c_j(x, y))$ for $j \in [n_e]$. For the multiple-expert case, the proof becomes more complex because $\text{argmax}_{j \in [n_e]} p_{n+j}$ may differ from $\text{argmax}_{j \in [n_e]} h(x, n + j)$. In contrast, these quantities coincide in the single-expert case. Moreover, the more complicated forms of both the surrogate loss and the deferral loss in the multiple-expert setting introduce further layers of complexity to the proof.

When $\mathcal{H} = \mathcal{H}_{\text{all}}$, the family of all measurable functions, the minimizability gaps vanish. In this case, Theorems 3.4 implies the excess error bounds and Bayes-consistency guarantees of $L_{\text{mae}}$. Along with Theorem 3.2, we conclude that $L_{\text{mae}}$ is both realizable $\mathcal{H}$-consistent and benefits from $\mathcal{H}$-consistency bounds (thus ensuring Bayes-consistency) for single-stage multiple-expert deferral.

## 4. Two-Stage Multiple-Expert Deferral

Recall that in the two-stage multiple-expert deferral setting, the goal is to find a deferral function $r : \mathcal{X} \times [n_e] \to \mathbb{R}$ that minimizes the following two-stage deferral loss:

$$
L_{\text{tdef}}(r, x, y) = \sum_{j=1}^{n_e} c_j(x, y) 1_{r(x)=j}.
$$

We denote by $\underline{c}_j \geq 0$ and $\overline{c}_j \geq 0$ the lower and upper bounds of the cost $c_j$, for any $j \in \{1, \ldots, n_e\}$.

### 4.1. Two Experts

We first consider the two-expert case ($n_e = 2$). The two-stage deferral loss can be then be written as follows:

$$
L_{\text{tdef}}(r, x, y) = c_1(x, y) 1_{r(x)=1} + c_2(x, y) 1_{r(x)=2}.
$$

A natural surrogate loss admits the following form:

$$
\begin{aligned}
L_\Phi(r, x, y) = c_1(x, y)\Phi(r(x, 2) - r(x, 1)) \\
+ c_2(x, y)\Phi(r(x, 1) - r(x, 2)), \quad (5)
\end{aligned}
$$

where $\Phi : \mathbb{R} \to \mathbb{R}_+$ is a decreasing function that defines a margin-based loss in binary classification. As an example, for $\Phi(t) = \log(1 + e^{-t})$, $L_\Phi$ can be expressed as the following cost-sensitive logistic loss:

$$
\begin{aligned}
L_\Phi(r, x, y) = c_1(x, y)\log\big(1 + e^{r(x,1)-r(x,2)}\big) \\
+ c_2(x, y)\log\big(1 + e^{r(x,2)-r(x,1)}\big). \quad (6)
\end{aligned}
$$

Next, we show that this surrogate loss is realizable $\mathcal{H}$-consistent with respect to $L_{\text{tdef}}$.

**Theorem 4.1.** *Assume that $\mathcal{R}$ is closed under scaling and that $\Phi$ satisfies $\lim_{t\to+\infty} \Phi(t) = 0$ and $\Phi(t) \geq 1_{t\leq 0}$. Then, the surrogate loss $L_\Phi$ is realizable $\mathcal{R}$-consistent with respect to $L_{\text{tdef}}$.*

The proof is provided in Appendix C.1. The following result further establishes that $L_\Phi$ satisfies an $\mathcal{R}$-consistency bound with respect to $L_{\text{tdef}}$, provided that $\Phi$ is associated with an $\mathcal{R}$-consistency bound in binary classification.

**Theorem 4.2.** *Assume that the following $\mathcal{R}$-consistency bound holds in binary classification:*

$$
\begin{aligned}
\mathcal{E}_{\ell_{0-1}}&(h) - \mathcal{E}_{\ell_{0-1}}(\mathcal{R}) + \mathcal{M}_{\ell_{0-1}}(\mathcal{R}) \\
&\leq \Gamma\big(\mathcal{E}_\Phi(h) - \mathcal{E}_\Phi(\mathcal{R}) + \mathcal{M}_\Phi(\mathcal{R})\big).
\end{aligned}
$$

*Then, the following $\mathcal{R}$-consistency bound holds in two-stage, two-expert deferral:*

$$
\begin{aligned}
\mathcal{E}_{L_{\text{tdef}}}&(r) - \mathcal{E}_{L_{\text{tdef}}}(\mathcal{R}) + \mathcal{M}_{L_{\text{tdef}}}(\mathcal{R}) \\
&\leq (\overline{c}_1 + \overline{c}_2)\Gamma\left( \frac{\mathcal{E}_{L_\Phi}(r) - \mathcal{E}_{L_\Phi}(\mathcal{R}) + \mathcal{M}_{L_\Phi}(\mathcal{R})}{\underline{c}_1 + \underline{c}_2} \right),
\end{aligned}
$$

*where constant factors $(\underline{c}_1 + \underline{c}_2)$ and $(\overline{c}_1 + \overline{c}_2)$ can be removed when $\Gamma$ is linear.*

The proof is presented in Appendix C.2. Note that, as shown by Awasthi et al. (2022a), when $\mathcal{H}$ consists of common function classes such as linear models, neural network families, or the family of all measurable functions, and for $\Phi(t) = e^{-t}$ or $\Phi(t) = \log(1 + e^{-t})$, $\Gamma$ can be chosen as the square-root function. Similarly, for $\Phi(t) = \max\{0, 1 - t\}$, $\Gamma$ can be chosen as the identity function, resulting in a linear bound. Combining this with Theorem 4.1, we conclude that $\mathsf{L}_\Phi$ is not only realizable $\mathcal{R}$-consistent but also benefits from $\mathcal{R}$-consistency bounds, thereby also ensuring Bayes-consistency, in the two-stage two-expert deferral setting.

### 4.2. Multiple Experts

Next, we consider the general multiple-expert case where $n_e > 2$. Motivated by the formulation of the surrogate loss in the two-expert case and the comp-sum losses (Mao et al., 2023f) in standard multi-class classification, we propose the following surrogate loss for the multiple-expert case:

$$\mathsf{L}_\Psi(r, x, y) \qquad (7)$$
$$= \sum_{j=1}^{n_e}\left(\sum_{j'\neq j} c_{j'}(x, y) - n_e + 2\right)\Psi\left(\frac{e^{r(x,j)}}{\sum_{j'\in[n_e]} e^{r(x,j')}}\right)$$

where $\Psi:[0, 1] \to \mathbb{R}_+ \cup \{+\infty\}$ is a decreasing function, such as $\Psi(u) = 1 - u$ or $\Psi(u) = -\log(u)$, as discussed in Section 3.1. Note that when $n_e = 2$, this formulation coincides with the one in the two-expert case, as shown in (5), with $\Phi: t \mapsto \Psi\left(\frac{e^t}{e^t+1}\right)$. As an example, in the case where $\Psi(u) = -\log(u)$, $\mathsf{L}_\Psi$ can be expressed as follows:

$$\sum_{j=1}^{n_e}\left(\sum_{j'\neq j} c_{j'}(x, y) - n_e + 2\right)\log\left(1 + \sum_{j'\neq j} e^{r(x,j')-r(x,j)}\right).$$

When $n_e = 2$, this formulation coincides with the one in the two-expert case, as shown in (6). Next, we show that the surrogate loss $\mathsf{L}_\Psi$ is realizable $\mathcal{H}$-consistent with respect to $\mathsf{L}_{\mathrm{tdef}}$ under natural assumptions about the cost functions and the auxiliary function $\Psi$.

**Theorem 4.3.** *Assume that $\mathcal{R}$ is closed under scaling and that $c_j(x, y) = 1_{g_j(x)\neq y}$, $j \in [n_e]$. Further, assume that for any $(x, y)$, there is at most one expert $j^* \in [n_e]$ for which $c_{j^*}(x, y) = 0$ and that $\Psi$ satisfies $\lim_{u\to 1^-}\Psi(u) = 0$. Then, the surrogate loss $\mathsf{L}_\Psi$ is realizable $\mathcal{R}$-consistent with respect to $\mathsf{L}_{\mathrm{tdef}}$.*

The proof is included in Appendix C.3. Next, we consider the case where $\Psi(u) = \Psi_q$, which satisfies the conditions $\lim_{u\to 1^-}\Psi(u) = 0$. The following result shows that $\mathsf{L}_{\Psi_q}$ also admits an $\mathcal{R}$-consistency bound with respect to $\mathsf{L}_{\mathrm{tdef}}$.

**Theorem 4.4.** *Assume that $\mathcal{R}$ is symmetric and complete and that the inequality $\sum_{j'\neq j} c_{j'}(x, y) \geq n_e - 2$ holds for all $j \in [n_e]$ and $(x, y) \in \mathcal{X} \times \mathcal{Y}$. Then, the following $\mathcal{R}$-*

*consistency bound holds:*

$$\mathcal{E}_{\mathsf{L}_{\mathrm{tdef}}}(r) - \mathcal{E}^*_{\mathsf{L}_{\mathrm{tdef}}}(\mathcal{R}) + \mathcal{M}_{\mathsf{L}_{\mathrm{tdef}}}(\mathcal{R})$$
$$\leq \Gamma\Big(\mathcal{E}_{\mathsf{L}_{\Psi_q}}(r) - \mathcal{E}^*_{\mathsf{L}_{\Psi_q}}(\mathcal{R}) + \mathcal{M}_{\mathsf{L}_{\Psi_q}}(\mathcal{R})\Big),$$

*where $\overline{C} = (n_e - 1)\max_{j\in[n_e]} \overline{c}_j - n_e + 2$, $\Gamma(t) = 2(\overline{C})^{\frac{1}{2}}\sqrt{t}$ when $q = 0$, $\Gamma(t) = 2(n_e^q)^{\frac{1}{2}}(\overline{C})^{\frac{1}{2}}\sqrt{t}$ when $q \in (0, 1)$, and $\Gamma(t) = n_e t$ when $q = 1$.*

Appendix C.4 contains the proof. Collectively, Theorems 4.3 and 4.4 show that for two-stage multiple-expert deferral, $\mathsf{L}_{\Psi_q}$ is both realizable $\mathcal{R}$-consistent, and admits $\mathcal{R}$-consistency bounds, thus ensuring Bayes-consistency.

## 5. Enhanced Bounds under Low-Noise Assumptions

In the previous sections, we established guarantees for realizable $\mathcal{H}$-consistency and $\mathcal{H}$-consistency bounds in the context of learning with multiple-expert deferral. Realizable $\mathcal{H}$-consistency provides guarantees under deterministic assumptions, while $\mathcal{H}$-consistency bounds extend to arbitrary distributions. This raises a natural question: can we offer guarantees for intermediate cases, particularly for distributions satisfying low-noise assumptions? In what follows, we address this question separately for the single-stage and two-stage scenarios.

### 5.1. Single-Stage Scenario

We first characterize the conditional error and conditional regret of the deferral loss. Let $y_{\max} = \arg\max_{y\in\mathcal{Y}} p(y|x)$, $p_{y_{\max}} = p(y_{\max}|x)$ and $p_{n+j} = \sum_{y\in\mathcal{Y}} p(y|x)(1 - c_j(x, y))$, $j \in [n_e]$. For any $h \in \mathcal{H}_{\mathrm{all}}$, define:

$$p_{\mathsf{h}(x)} = \begin{cases} p(\mathsf{h}(x)|x) & \mathsf{h}(x) \in [n] \\ p_{n+j} & \mathsf{h}(x) = n + j. \end{cases}$$

**Lemma 5.1.** *Assume that $\mathcal{H}$ is symmetric and complete. Then, for any $h \in \mathcal{H}$ and input $x \in \mathcal{X}$, the conditional error and conditional regret of the deferral loss are given by:*

$$\mathcal{C}_{\mathsf{L}_{\mathrm{def}}}(h, x) = 1 - p_{\mathsf{h}(x)}$$
$$\Delta\mathcal{C}^*_{\mathsf{L}_{\mathrm{def}},\mathcal{H}}(h, x) = \max\left\{p_{y_{\max}}, \max_{j\in[n_e]} p_{n+j}\right\} - p_{\mathsf{h}(x)}.$$

The proof of Lemma 5.1 is included in Appendix B.3.1.

The original Tsybakov noise definition was introduced and further studied in the context of standard classification (Mammen & Tsybakov, 1999; Mao et al., 2024e). We extend this definition to the setting of the deferral loss with multiple experts and establish new $\mathcal{H}$-consistency guarantees under this noise assumption.

This extension relies on the concept of the *Bayes classifier for the deferral loss*. By (Mao et al., 2024d, Lemma 2.1), there exists a measurable function $h^*$ that satisfies $p_{h^*(x)} = \max\{p_{y_{\max}}, \max_{j \in [n_e]} p_{n+j}\}$, for all $x \in \mathcal{X}$ which identifies $h^*$ as the Bayes classifier for the deferral loss. We can now define the *minimal margin for a point $x \in \mathcal{X}$* as $\gamma(x) = p_{h^*(x)} - \sup_{y \neq h^*(x)} p_y$ and the *single-stage deferral Tsybakov noise assumption* as follows: there exist $B > 0$ and $\alpha \in [0, 1)$ such that the following inequality holds:

$$\forall t > 0, \quad \mathbb{P}[\gamma(X) \leq t] \leq B t^{\frac{\alpha}{1-\alpha}}. \tag{8}$$

Thus, this noise assumption posits that the probability of observing a small margin is relatively low. Note that when $\alpha \to 1$, $t^{\frac{\alpha}{1-\alpha}} \to 0$, which corresponds to an analogue of Massart's noise assumption in the deferral setting. The following lemma establishes basic properties of single-stage deferral Tsybakov noise assumption, extending a similar result from the classification setting by Mao et al. (2024e).

**Lemma 5.2.** *The single-stage deferral Tsybakov noise assumption implies that there exists a constant $c = \frac{B^{1-\alpha}}{\alpha^\alpha} > 0$ such that the following inequalities hold for any $h \in \mathcal{H}$:*

$$\mathbb{E}[1_{h(x) \neq h^*(x)}] \leq c \, \mathbb{E}[\gamma(X) 1_{h(x) \neq h^*(x)}]^\alpha$$
$$\leq c [\mathcal{E}_{\mathsf{L}_{\text{def}}}(h) - \mathcal{E}_{\mathsf{L}_{\text{def}}}(h^*)]^\alpha.$$

The proof of Lemma 5.2 can be found in Appendix D.1.1. We first derive an $\mathcal{H}$-consistency bound in terms of the quantity $1_{h(x) \neq h^*(x)}$, which appears on the left-hand side of the statement of Lemma 5.2.

**Theorem 5.3.** *Consider the setting of single-stage multiple-expert deferral. Assume that the following holds for all $h \in \mathcal{H}$ and $x \in \mathcal{X}$: $\Delta\mathcal{C}_{\mathsf{L}_{\text{def}},\mathcal{H}}(h, x) \leq \Gamma(\Delta\mathcal{C}_{\mathsf{L},\mathcal{H}}(h, x))$, with $\Gamma(x) = x^{\frac{1}{s}}$, for some $s \geq 1$ with conjugate number $t \geq 1$, that is $\frac{1}{s} + \frac{1}{t} = 1$. Then, for any $h \in \mathcal{H}$,*

$$\mathcal{E}_{\mathsf{L}_{\text{def}}}(h) - \mathcal{E}_{\mathsf{L}_{\text{def}}}^*(\mathcal{H}) + \mathcal{M}_{\mathsf{L}_{\text{def}}}(\mathcal{H})$$
$$\leq \mathbb{E}_X[1_{h(X) \neq h^*(X)}]^{\frac{1}{t}} (\mathcal{E}_{\mathsf{L}}(h) - \mathcal{E}_{\mathsf{L}}^*(\mathcal{H}) + \mathcal{M}_{\mathsf{L}}(\mathcal{H}))^{\frac{1}{s}}.$$

The proof of Theorem 5.3 is included in Appendix D.1.2. Theorem 5.3 offers a stronger theoretical guarantee compared to standard $\mathcal{H}$-consistency bounds, under the assumption $\Delta\mathcal{C}_{\mathsf{L}_{\text{def}},\mathcal{H}}(h, x) \leq \Gamma(\Delta\mathcal{C}_{\mathsf{L},\mathcal{H}}(h, x))$. This bound includes the hypothesis-dependent factor $\mathbb{E}_X[1_{h(X) \neq h^*(X)}]^{\frac{1}{t}} \leq 1$, which is more favorable than the constant factor of one in standard bounds.

Next, we assume that the Tsybakov noise assumption holds and also that there is no approximation error, that is $\mathcal{M}_{\mathsf{L}_{\text{def}}}(\mathcal{H}) = 0$.

**Theorem 5.4.** *Consider the setting of single-stage multiple-expert deferral where the Tsybakov noise assumption holds, and no approximation error occurs, that is, $\mathcal{E}_{\mathsf{L}_{\text{def}}}^*(\mathcal{H}) =$*

$\mathcal{E}_{\mathsf{L}_{\text{def}}}^*(\mathcal{H}_{\text{all}})$. *Assume that the following holds for all $h \in \mathcal{H}$ and $x \in \mathcal{X}$: $\Delta\mathcal{C}_{\mathsf{L}_{\text{def}},\mathcal{H}}(h, x) \leq \Gamma(\Delta\mathcal{C}_{\mathsf{L},\mathcal{H}}(h, x))$, with $\Gamma(x) = x^{\frac{1}{s}}$, for some $s \geq 1$. Then, for any $h \in \mathcal{H}$,*

$$\mathcal{E}_{\mathsf{L}_{\text{def}}}(h) - \mathcal{E}_{\mathsf{L}_{\text{def}}}^*(\mathcal{H})$$
$$\leq c^{\frac{s-1}{s-\alpha(s-1)}} [\mathcal{E}_{\mathsf{L}}(h) - \mathcal{E}_{\mathsf{L}}^*(\mathcal{H}) + \mathcal{M}_{\mathsf{L}}(\mathcal{H})]^{\frac{1}{s-\alpha(s-1)}}.$$

The proof is given in Appendix D.1.3. Note that as $\alpha \to 1$, where the Tsybakov noise corresponds to Massart's noise assumption, this result yields an $\mathcal{H}$-consistency bound with a linear functional form, improving upon the standard $\mathcal{H}$-consistency bounds, which have the functional form $\Gamma(x) = x^{\frac{1}{s}}$, when such bounds exist. This shows that for any smooth surrogate losses with an $\mathcal{H}$-consistency bound, under Massart's noise assumption, we can obtain $\mathcal{H}$-consistency bounds as favorable as those of $\mathsf{L} = \mathsf{L}_{\text{mae}}$, which always admit a linear dependency, as shown in Theorem 3.4.

For example, in the special case of a single-expert, Mao et al. (2024i, Theorem 4.2) shows that square-root $\mathcal{H}$-consistency bounds hold ($s = \frac{1}{2}$) for $\mathsf{L} = \mathsf{L}_{\Psi_q}$ with $q \in [0, 1)$. Theorem 5.4 provides refined $\mathcal{H}$-consistency bounds for these surrogate losses: linear dependence when Massart's noise assumption holds ($\alpha \to 1$) and an intermediate rate between linear and square-root for other values of $\alpha \in (0, 1)$.

Furthermore, we know that the realizability assumption can be viewed as a special case of Massart's noise assumption. Thus, Theorem 5.4 provides intermediate guarantees between realizable $\mathcal{H}$-consistency and $\mathcal{H}$-consistency bounds for single-stage deferral surrogate losses, such as $\mathsf{L}_{\text{L2D}}$ considered in (Mao et al., 2024i).

### 5.2. Two-Stage Scenario

We now extend our results to the two-stage scenario. As in the single-stage case, we begin by characterizing the best-in-class conditional error and the conditional regret of the two-stage deferral loss function $\mathsf{L}_{\text{tdef}}$ (Lemma C.1 in Appendix C.4.1). In particular, the Bayes conditional error is $\mathcal{C}_{\mathsf{L}_{\text{tdef}}}^*(\mathcal{R}_{\text{all}}, x) = \min_{j \in [n_e]} \sum_{y \in \mathcal{Y}} p(y|x) c_j(x, y)$.

By (Mao et al., 2024d, Lemma 2.1), there exists a measurable function $r^*$ such that

$$\sum_{y \in \mathcal{Y}} p(y|x) c_{r^*(x)}(x, y) = \min_{j \in [n_e]} \sum_{y \in \mathcal{Y}} p(y|x) c_j(x, y),$$

for all $x \in \mathcal{X}$. This function $r^*$ thus defines a Bayes classifier for the two-stage deferral loss. Define the *minimal margin for a point $x \in \mathcal{X}$* as follows: $\gamma(x) =$

$$\inf_{j \neq r^*(x)} \sum_{y \in \mathcal{Y}} p(y|x) c_j(x, y) - \sum_{y \in \mathcal{Y}} p(y|x) c_{r^*(x)}(x, y) \geq 0.$$

The *two-stage Tsybakov noise assumption* can then be defined as follows: there exist $B > 0$ and $\alpha \in [0, 1)$ such that

the following inequality holds:

$$\forall t > 0, \quad \mathbb{P}[\gamma(X) \leq t] \leq B t^{\frac{\alpha}{1-\alpha}}. \tag{9}$$

As in the single-stage setting, Lemma D.2 in Appendix D.2.1 establishes fundamental properties of the Tsybakov noise assumption within the context of the two-stage scenario. We also derive the enhanced bounds (Theorems D.3 and D.4) for the two-stage scenario, with their detailed presentation deferred to Appendix D.2.2 due to space limitations.

# 6. Experiments

In this section, we evaluate the empirical performance of our proposed surrogate loss, comparing it against existing baselines in both single-stage and two-stage learning-to-defer scenarios with multiple experts. Our objective is to confirm that our loss functions match the best-known results in non-realizable settings for both single- and two-stage cases while outperforming them in realizable settings, as predicted by our theoretical analysis.

**Experimental setup**. For our experiments, we used four widely used datasets: CIFAR-10, CIFAR-100 (Krizhevsky, 2009), SVHN (Netzer et al., 2011), and Tiny ImageNet (Le & Yang, 2015). Each dataset was trained for 200 epochs. We adopted ResNet-16 (He et al., 2016) for the predictor/deferral models. The Adam optimizer (Kingma & Ba, 2015) was used with a weight decay of $1 \times 10^{-3}$ and a batch size of 1024. Following (Mozannar et al., 2023; Mao et al., 2024a), we define the cost function as the expert's classification error: $c_j(x, y) = 1_{g_j(x) \neq y}$ in our empirical analysis. We reported the means and standard deviations over five runs. For simplicity, we omitted the standard deviations of the deferral ratios.

**Single-stage multiple-expert deferral**. In the single-stage scenario, we compared our proposed surrogate $\mathsf{L}_\Psi$ (3) with $\Psi(u) = 1 - u$ with the baseline surrogate losses proposed in (Verma et al., 2023) and (Mao et al., 2024a). While these surrogate losses are Bayes-consistent, they are not realizable $\mathcal{H}$-consistent. For both surrogate losses, we used the logistic loss function as their auxiliary function in our experiments. Building on the setup described in (Mao et al., 2024a), we used two experts, each with a distinct domain of expertise. Expert 1 always predicts from the first 30% of classes and is correct whenever the true label falls within this range. Similarly, Expert 2 predicts from the next 30% of classes, following the same criterion for accuracy. As shown in Table 2, our method achieves comparable *single-stage system accuracy (SA)*, defined as the average value of $[1 - \mathsf{L}_{\mathrm{def}}(h, x, y)]$ on the test data, to the baselines. Due to the specific realizable form of our surrogate losses, it also defers significantly more and exhibits a greater tendency to choose expert 2 compared to the baselines. This highlights a key effect of our proposed surrogate: it induces deferral

*Table 2.* Comparison of Our Method with Single-Stage Baselines.

| Method | Dataset | SA (%) | Ratio of Deferral (%) | | |
|---|---|---|---|---|---|
| | | | PRED | EXP 1 | EXP 2 |
| Verma et al. (2023) | | 91.85 ±0.08 | 62.88 | 11.23 | 25.89 |
| Mao et al. (2024a) | CIFAR-10 | 91.98 ± 0.09 | 66.63 | 13.65 | 19.72 |
| Ours | | 92.21 ± 0.18 | 38.79 | 30.45 | 30.76 |
| Verma et al. (2023) | | 64.13 ± 0.15 | 39.77 | 35.95 | 24.28 |
| Mao et al. (2024a) | CIFAR-100 | 64.93 ± 0.22 | 49.19 | 27.69 | 23.12 |
| Ours | | 65.39 ± 0.31 | 34.38 | 30.52 | 35.10 |
| Verma et al. (2023) | | 95.91 ± 0.08 | 64.66 | 25.96 | 9.38 |
| Mao et al. (2024a) | SVHN | 95.78 ± 0.06 | 78.93 | 10.09 | 10.98 |
| Ours | | 96.08 ± 0.11 | 27.11 | 43.42 | 29.47 |
| Verma et al. (2023) | | 50.61 ± 0.50 | 46.96 | 25.59 | 27.45 |
| Mao et al. (2024a) | Tiny ImageNet | 50.89 ± 0.49 | 45.56 | 37.88 | 16.56 |
| Ours | | 51.13 ± 0.56 | 21.68 | 32.60 | 45.72 |

*Table 3.* Comparison of Our Method with Two-Stage Baselines.

| Method | Dataset | SA (%) | Ratio of Deferral (%) | | |
|---|---|---|---|---|---|
| | | | EXP 1 | EXP 2 | EXP 3 |
| Mao et al. (2023a) | | 92.92 ± 0.14 | 30.23 | 30.49 | 39.28 |
| Ours | CIFAR-10 | 93.34 ± 0.18 | 29.34 | 30.90 | 39.76 |
| Mao et al. (2023a) | | 67.99 ± 0.19 | 32.44 | 34.42 | 33.14 |
| Ours | CIFAR-100 | 68.70 ± 0.23 | 31.34 | 29.89 | 38.77 |
| Mao et al. (2023a) | | 96.30 ± 0.06 | 42.38 | 29.68 | 27.94 |
| Ours | SVHN | 96.47 ± 0.08 | 42.52 | 29.70 | 27.78 |
| Mao et al. (2023a) | | 45.81 ± 0.17 | 36.64 | 28.64 | 34.72 |
| Ours | Tiny ImageNet | 46.62 ± 0.24 | 26.82 | 38.15 | 35.03 |

behavior that is qualitatively different from prior approaches, particularly on the SVHN and Tiny ImageNet datasets.

**Two-stage multiple-expert deferral**. In the two-stage scenario, we compared our proposed surrogate loss $\mathsf{L}_\Psi$ (7) with $\Psi(u) = -\log(u)$ with the baseline surrogate losses proposed in (Mao et al., 2023a). While the baseline surrogate losses are Bayes-consistent, they are not realizable $\mathcal{H}$-consistent with cost functions based on classification error: $c_j(x, y) = 1_{g_j(x) \neq y}$. In our experiments, we used the multinomial logistic loss as the multi-class classification surrogate for the baseline surrogate losses. As in the single-stage scenario, we considered a setting where multiple experts are available, each with a clear domain of expertise. We used three experts, with two of them the same as those adopted in the single-stage setting. Additionally, we introduced a third expert that predicts from the remaining 40% of classes and is accurate if the true label is within this range. As shown in Table 3, our method achieves comparable *two-stage system accuracy (SA)*, defined as the average value of $[1 - \mathsf{L}_{\mathrm{tdef}}(h, x, y)]$ on the test data, to the baselines. The different deferral ratios it achieves, compared to the baseline, demonstrate the effect of our proposed surrogate loss.

**Realizable multiple-expert deferral**. We also conducted an additional experiment in the realizable scenario, by extending the synthetic Mixture-of-Gaussians dataset from (Mozannar et al., 2023) to the multiple-expert setting. This

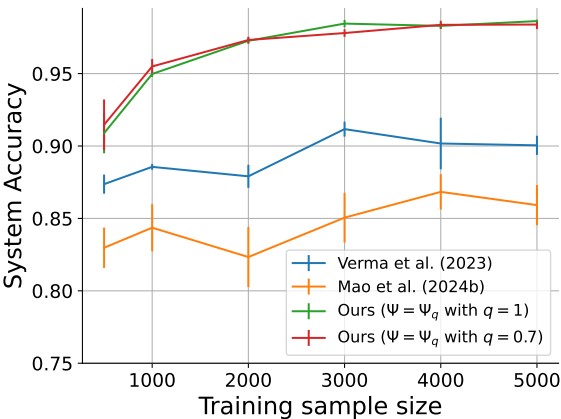

*Figure 1.* System Accuracy vs. Training Sample Size.

extended dataset is realizable by linear functions, as there exists a randomly selected linear hypothesis $h^* \in \mathcal{H}$ that achieves zero single-stage deferral loss, $\mathcal{E}_{\mathsf{L}_{\mathrm{def}}}(h^*) = 0$. We set the true label to match the predictor's prediction when deferral does not occur according to $h^*$. Otherwise, the true label is generated uniformly at random. We used two experts whose predictions align with the true label only when they are chosen by $h^*$ to make the prediction. For our surrogate loss $\mathsf{L}_\Psi$ (3), we selected $\Psi = \Psi_q$ with $q = 0.7$ and $q = 1$. We compared these surrogate losses to those introduced in (Verma et al., 2023) and (Mao et al., 2024a). Figure 1 shows system accuracy as a function of training samples on the realizable Mixture-of-Gaussians distribution, verifying our theoretical results that both choices of $\mathsf{L}_\Psi$ are realizable $\mathcal{H}$-consistent, while the two baselines are not.

The behavior for lower values of $q$ is similar to that observed for $q = 0.7$ and $q = 1$. Note that for any $q > 0$, Theorem 3.2 guarantees realizable $\mathcal{H}$-consistency. We chose $q = 0.7$ and $q = 1$ in Figure 1 because they correspond to standard choices in prior work, $q = 0.7$ is a commonly suggested value within the generalized cross-entropy family (e.g., (Zhang & Sabuncu, 2018); (Mao et al., 2024a)), and $q = 1$ corresponds to the mean absolute error loss, as used in ((Ghosh et al., 2017); (Mao et al., 2024a)). In this realizable distribution, we observe that our proposed surrogate losses achieve close to $100\%$ system accuracy, while all other baselines fail to find a near-zero-error solution. This demonstrates that our surrogates are realizable $\mathcal{H}$-consistent, whereas the compared baselines are not.

Note that the goal of our experiments on non-realizable data is not to claim that our proposed surrogate losses outperform the baselines (Verma et al., 2023; Mao et al., 2024a; 2023a) in such settings. Rather, our results show that our surrogate losses match the performance of these state-of-the-art baselines in non-realizable settings in terms of system accuracy, as both are supported by consistency bounds. The strength of our approach lies in the realizable setting, where our surrogate losses are guaranteed to be realizable $\mathcal{H}$-consistent,

unlike the baselines, as illustrated in Figure 1.

# 7. Future Work

Our paper primarily focuses on theoretically principled surrogate losses for learning to defer with multiple experts, grounded in realizable $\mathcal{H}$-consistency and $\mathcal{H}$-consistency bounds, within the learning theory framework. While we have empirically shown that minimizing our proposed loss functions outperforms the baselines in realizable settings and matches the best-known results in synthetic non-realizable settings, as predicted by our theoretical analysis, we recognize the importance of further empirical exploration. Accordingly, we plan to dedicate future work to conducting a more extensive empirical analysis in real-world settings, particularly in important scenarios where expert domains overlap and the data is heterogeneous.

We also acknowledge the relevance of more realistic and challenging scenarios, such as those where expert predictions are unavailable during training. Extending our methods to these scenarios is a promising direction for future work. For example, it would be interesting to adapt our framework to handle previously unseen experts at test time, as studied by Tailor et al. (2024) and to incorporate post-processing frameworks for learning to defer under constraints such as OOD detection and long-tail classification, as explored by Charusaie & Samadi (2024).

Our deferral framework also offers promising applications in the context of large language models (LLMs). Given the significant resources required to retrain LLMs, a two-stage method is a practical and scalable solution (Mao et al., 2023a). This does not require modifying LLMs or the loss function used to train them. Our proposed two-stage surrogate loss is particularly well-suited for use with pre-trained LLMs. In the presence of multiple LLMs (or experts), using our method, uncertain predictions can be deferred to more specialized or domain-specific pre-trained models, thereby both improving the reliability and accuracy of the overall system and improving efficiency.

# 8. Conclusion

We presented novel surrogate losses functions and algorithms for multiple-expert deferral, offering strong theoretical guarantees for both single- and two-stage scenarios. Our results, including realizable $\mathcal{H}$-consistency, $\mathcal{H}$-consistency bounds, and Bayes-consistency, extend to multiple-expert settings and provide enhanced guarantees under low-noise assumptions. Our analysis can be leveraged to study and design algorithms for other routing problems, for example when incorporating constraints, such as limiting the frequency with which an expert can be used. Experimental results confirm the effectiveness of our principled techniques.

## Acknowledgements

We thank the anonymous reviewers for their valuable feedback and constructive suggestions.

## Impact Statement

This paper presents work whose goal is to advance the field of Machine Learning. There are many potential societal consequences of our work, none which we feel must be specifically highlighted here.

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

# Contents of Appendix

# A. Related work

Single-stage learning to defer, where a predictor and a deferral function are jointly trained, was pioneered by Cortes, DeSalvo, and Mohri (2016a;b; 2023) and further developed in subsequent work. This includes studies on abstention with constant costs (Charoenphakdee et al., 2021; Cao et al., 2022; Li et al., 2023; Cheng et al., 2023; Mao et al., 2024c;b; Mohri et al., 2024; Narasimhan et al., 2024) and deferral with instance- and label-dependent costs (Mozannar & Sontag, 2020; Verma & Nalisnick, 2022; Mozannar et al., 2023; Verma et al., 2023; Cao et al., 2023; Mao et al., 2024a; Wei et al., 2024; Mao et al., 2024i). In this paradigm, the deferral function decides how input instances should be optimally assigned to the most suitable expert from a diverse set. This method has demonstrated advantages over *confidence-based* approaches, which rely solely on the predictor's output magnitude (Chow, 1957; 1970; Bartlett & Wegkamp, 2008; Yuan & Wegkamp, 2010; 2011; Ramaswamy et al., 2018; Ni et al., 2019; Jitkrittum et al., 2023); and *selective classification* methods, which use a fixed selection rate and cannot incorporate expert-modeled cost functions (El-Yaniv et al., 2010; El-Yaniv & Wiener, 2012; Wiener & El-Yaniv, 2011; 2012; 2015; Geifman & El-Yaniv, 2017; 2019; Acar et al., 2020; Gangrade et al., 2021; Zaoui et al., 2020; Jiang et al., 2020; Shah et al., 2022).

The *learning to defer* (L2D) problem, which incorporates human expert decisions into the cost function, was introduced by Madras et al. (2018) and further studied by Raghu et al. (2019); Wilder et al. (2021); Pradier et al. (2021); Mozannar & Sontag (2020); Verma & Nalisnick (2022); Charusaie et al. (2022); Mozannar et al. (2023); Mao et al. (2023a; 2024a;i). It is typically formulated using a deferral loss function that incorporates instance-specific costs associated with each expert. Directing optimizing this loss function is intractable for the hypothesis sets commonly used in applications. Thus, learning-to-defer algorithms rely on optimizing a surrogate loss function instead, that serves as a proxy for the original target loss function. Yet, what guarantees can we rely on when optimizing such surrogate loss functions?

This question, which involves analyzing the consistency guarantees of surrogate losses with respect to the deferral loss, has been studied under two main scenarios (Mao, 2025): the *single-stage* scenario, where a predictor and a deferral function are jointly learned (Mozannar & Sontag, 2020; Verma & Nalisnick, 2022; Charusaie et al., 2022; Mozannar et al., 2023; Mao et al., 2024a), and a *two-stage* scenario, where the predictor is pre-trained and fixed as an expert, and while only the deferral function is subsequently learned (Mao et al., 2023a).

In particular, Mozannar & Sontag (2020), Verma & Nalisnick (2022), and Charusaie et al. (2022) proposed surrogate loss functions for the single-stage *single-expert* case by generalizing the cross-entropy loss, the one-versus-all loss, and, more generally, a broad family of surrogate losses for multi-class classification in the context of learning to defer. However, Mozannar et al. (2023) later showed that these surrogate loss functions do not satisfy *realizable $\mathcal{H}$-consistency*. They suggested an alternative surrogate loss that achieves this property but left open the question of whether it was also Bayes-consistent. This was later resolved by Mao et al. (2024i), who introduced a broader family of surrogate losses that simultaneously achieve Bayes-consistency, realizable $\mathcal{H}$-consistency, and $\mathcal{H}$-*consistency bounds*. For the single-stage *multiple-expert* (Hemmer et al., 2022; Keswani et al., 2021; Kerrigan et al., 2021; Straitouri et al., 2022; Benz & Rodriguez, 2022) case, Verma et al. (2023) were the first to extend the surrogate loss proposed in (Verma & Nalisnick, 2022) and (Mozannar & Sontag, 2020) to accommodate multiple experts. Building on this, Mao et al. (2024a) further generalized the surrogate loss from (Mozannar & Sontag, 2020), introducing a broader family of surrogate losses tailored to the multiple-expert case. Furthermore, Mao et al. (2024a) proved that their surrogate losses benefit from $\mathcal{H}$-consistency bounds in the multiple-expert case, thereby ensuring Bayes-consistency. However, these loss functions are not realizable $\mathcal{H}$-consistent even in the single-expert case, as they are extensions of the earlier loss functions. In the *two-stage* scenario, Mao et al. (2023a) introduced surrogate losses that are Bayes-consistent and realizable $\mathcal{H}$-consistent for constant costs. However, their realizable $\mathcal{H}$-consistency does not extend to cost functions of interest, which are based on classification error. Other extensions include single-stage and two-stage multiple-expert deferral in regression (Mao et al., 2024h), single-stage learning to defer to a population (Tailor et al., 2024), and two-stage multi-expert deferral in multi-task learning (Montreuil et al., 2025d), adversarial robustness (Montreuil et al., 2025a), query optimization and allocation (Montreuil et al., 2024), and top-$k$ prediction (Montreuil et al., 2025b;c).

Further research has explored post-hoc methods. Okati et al. (2021) proposed an alternative optimization approach for the predictor and rejector, while Narasimhan et al. (2022) offered corrections for underfitting surrogate losses (Liu et al., 2024). Charusaie & Samadi (2024) developed a unified post-processing framework for multi-objective L2D based on a generalized Neyman-Pearson Lemma (Neyman & Pearson, 1933). Additionally, Cao et al. (2023) introduced an asymmetric softmax function for deriving valid probability estimates in L2D. Wei et al. (2024) explored dependent Bayes optimality, which elucidates the dependencies involved in deferral decisions within the L2D framework. The L2D framework and its variants

have been applied in various domains, including regression, multi-task learning, adversarial robustness, top-$k$ prediction, query optimization and allocation, human-in-the-loop systems and reinforcement learning (De et al., 2020; 2021; Zhao et al., 2021; Mozannar et al., 2022; Straitouri et al., 2021; Joshi et al., 2023; Gao et al., 2021; Hemmer et al., 2023; Chen et al., 2024; Palomba et al., 2024).

## B. Single-stage multiple-expert deferral: proofs.

### B.1. Proof of Lemma 3.1

**Lemma 3.1.** *The deferral loss can be expressed as follows:* $\forall (h, x, y) \in \mathcal{H}_{\mathrm{all}} \times \mathcal{X} \times \mathcal{Y}$,

$$\mathsf{L}_{\mathrm{def}}(h, x, y) = \left[\sum_{j=1}^{n_e} c_j(x, y) + 1 - n_e\right] 1_{\mathsf{h}(x) \neq y} + \sum_{j=1}^{n_e} \left[1 - c_j(x, y)\right] 1_{\mathsf{h}(x) \neq y} 1_{\mathsf{h}(x) \neq n+j}.$$

*Proof.* Observe that for any $(x, y) \in \mathcal{X} \times \mathcal{Y}$, we have $1_{\mathsf{h}(x) = n+j} = 1_{\mathsf{h}(x) \neq y} 1_{\mathsf{h}(x) = n+j}$, since $\mathsf{h}(x) = n + j$ implies $\mathsf{h}(x) \neq y$. Thus, using additionally $1_{\mathsf{h}(x) \in [n]} = \prod_{j=1}^{n_e} 1_{\mathsf{h}(x) \neq n+j}$, the deferral loss can be rewritten as follows for all $(x, y) \in \mathcal{X} \times \mathcal{Y}$:

$$\mathsf{L}_{\mathrm{def}}(h, x, y) = 1_{\mathsf{h}(x) \neq y} 1_{\mathsf{h}(x) \in [n]} + \sum_{j=1}^{n_e} c_j(x, y) 1_{\mathsf{h}(x) = n+j}$$

$$= 1_{\mathsf{h}(x) \neq y} \prod_{j=1}^{n_e} 1_{\mathsf{h}(x) \neq n+j} + \sum_{j=1}^{n_e} c_j(x, y) 1_{\mathsf{h}(x) \neq y} 1_{\mathsf{h}(x) = n+j}$$

$$= 1_{\mathsf{h}(x) \neq y} \prod_{j=1}^{n_e} 1_{\mathsf{h}(x) \neq n+j} + \sum_{j=1}^{n_e} c_j(x, y) 1_{\mathsf{h}(x) \neq y} (1 - 1_{\mathsf{h}(x) \neq n+j})$$

$$= \sum_{j=1}^{n_e} c_j(x, y) 1_{\mathsf{h}(x) \neq y} + 1_{\mathsf{h}(x) \neq y} \prod_{j=1}^{n_e} 1_{\mathsf{h}(x) \neq n+j} - \sum_{j=1}^{n_e} c_j(x, y) 1_{\mathsf{h}(x) \neq y} 1_{\mathsf{h}(x) \neq n+j}$$

$$= \sum_{j=1}^{n_e} c_j(x, y) 1_{\mathsf{h}(x) \neq y} + \sum_{j=1}^{n_e} \left[1 - c_j(x, y)\right] 1_{\mathsf{h}(x) \neq y} 1_{\mathsf{h}(x) \neq n+j} + 1_{\mathsf{h}(x) \neq y} \left[\prod_{j=1}^{n_e} 1_{\mathsf{h}(x) \neq n+j} - \sum_{j=1}^{n_e} 1_{\mathsf{h}(x) \neq n+j}\right]$$
$$\text{(add and subtract } \textstyle\sum_{j=1}^{n_e} 1_{\mathsf{h}(x) \neq y} 1_{\mathsf{h}(x) \neq n+j})$$

$$= \sum_{j=1}^{n_e} c_j(x, y) 1_{\mathsf{h}(x) \neq y} + \sum_{j=1}^{n_e} \left[1 - c_j(x, y)\right] 1_{\mathsf{h}(x) \neq y} 1_{\mathsf{h}(x) \neq n+j} + 1_{\mathsf{h}(x) \neq y} (1 - n_e)$$

$$(\textstyle\prod_{j=1}^{n_e} 1_{\mathsf{h}(x) \neq n+j} = \begin{cases} 1 & \mathsf{h}(x) \in [n] \\ 0 & \text{otherwise} \end{cases} \text{ and } \textstyle\sum_{j=1}^{n_e} 1_{\mathsf{h}(x) \neq n+j} = \begin{cases} n_e & \mathsf{h}(x) \in [n] \\ n_e - 1 & \text{otherwise} \end{cases})$$

$$= \left[\sum_{j=1}^{n_e} c_j(x, y) + 1 - n_e\right] 1_{\mathsf{h}(x) \neq y} + \sum_{j=1}^{n_e} \left[1 - c_j(x, y)\right] 1_{\mathsf{h}(x) \neq y} 1_{\mathsf{h}(x) \neq n+j}.$$

This completes the proof. $\qquad\square$

### B.2. Proof of Theorem 3.2

**Theorem 3.2.** *Assume $n_e \geq 2$ and $\mathcal{H}$ closed under scaling. Then, if $\Psi$ satisfies $\lim_{u \to 1^-} \Psi(u) = 0$ and $\lim_{u \to 0^+} \Psi(u) = 1$, then the surrogate loss $\mathsf{L}_\Psi$ is realizable $\mathcal{H}$-consistent with respect to $\mathsf{L}_{\mathrm{def}}$.*

*Proof.* Let $h^*$ be a best-in-class predictor such that $\mathcal{E}_{\mathsf{L}_{\mathrm{def}}}(h^*) = 0$. Note that

$$\mathcal{E}^*_{\mathsf{L}}(\mathcal{H}) \leq \lim_{\alpha \to +\infty} \mathcal{E}_{\mathsf{L}}(\alpha h^*)$$

$$= \lim_{\alpha \to +\infty} \mathbb{E}[\mathsf{L}(\alpha h^*, x, y) \mid h^*(x) \in [n]] \mathbb{P}(h^*(x) \in [n])$$

$$+ \sum_{k=1}^{n_e} \lim_{\alpha \to +\infty} \mathbb{E}[\mathsf{L}(\alpha h^*, x, y) \mid h^*(x) = n+k] \mathbb{P}(h^*(x) = n+k)$$

If $h^*(x) \in [n]$, then, we must have $h^*(x) = y$ for any $(x, y) \in \mathcal{X} \times \mathcal{Y}$. In this case,

$$\lim_{\alpha \to +\infty} \mathbb{E}[\mathsf{L}(\alpha h^*, x, y) \mid h^*(x) \in [n]]$$

$$\leq \mathbb{E}\left[\left[\sum_{j=1}^{n_e} c_j(x, y) + 1 - n_e\right] \lim_{\alpha \to +\infty} \Psi\left(\frac{e^{\alpha h^*(x,y)}}{\sum_{y' \in \overline{y}} e^{\alpha h^*(x,y')}}\right)\right]$$

$$+ \mathbb{E}\left[\sum_{j=1}^{n_e} [1 - c_j(x, y)] \lim_{\alpha \to +\infty} \Psi\left(\frac{e^{\alpha h^*(x,y)} + e^{\alpha h^*(x,n+j)}}{\sum_{y' \in \overline{y}} e^{\alpha h^*(x,y')}}\right)\right]$$

$$= \mathbb{E}[0] + \mathbb{E}[0]$$

$$= 0.$$

If $h^*(x) = n + k$, then, we must have $c_k(x, y) = 0$ for any $(x, y) \in \mathcal{X} \times \mathcal{Y}$. In this case,

$$\lim_{\alpha \to +\infty} \mathbb{E}[\mathsf{L}(\alpha h^*, x, y) \mid h^*(x) = n + k]$$

$$\leq \mathbb{E}\left[\left[\sum_{j=1}^{n_e} c_j(x, y) + 1 - n_e\right] \lim_{\alpha \to +\infty} \Psi\left(\frac{e^{\alpha h^*(x,y)}}{\sum_{y' \in \overline{y}} e^{\alpha h^*(x,y')}}\right)\right]$$

$$+ \mathbb{E}\left[\sum_{j=1}^{n_e} [1 - c_j(x, y)] \lim_{\alpha \to +\infty} \Psi\left(\frac{e^{\alpha h^*(x,y)} + e^{\alpha h^*(x,n+j)}}{\sum_{y' \in \overline{y}} e^{\alpha h^*(x,y')}}\right)\right]$$

$$\leq \mathbb{E}\left[\sum_{j \neq k} c_j(x, y) + 1 - n_e\right] + \mathbb{E}\left[\sum_{j \neq k} [1 - c_j(x, y)]\right]$$

$$= 0.$$

Therefore, we have $\mathcal{E}_\mathsf{L}^*(\mathcal{H}) = 0$. $\qquad \square$

## B.3. Proof of $\mathcal{H}$-consistency bounds

### B.3.1. Simplified notation and proof of Lemma 5.1

Let $p(y|x) = \mathbb{P}(Y = y \mid X = x)$ be the conditional probability of $Y = y$ given $X = x$. Let $y_{\max} = \operatorname{argmax}_{y \in \mathcal{Y}} p(y|x)$, $p_{y_{\max}} = p(y_{\max}|x)$ and $p_{n+j} = \sum_{y \in \mathcal{Y}} p(y|x)(1 - c_j(x, y))$, $j \in [n_e]$. We denote by $p_{h(x)} = \begin{cases} p(h(x)|x) & h(x) \in [n] \\ p_{n+j} & h(x) = n + j. \end{cases}$ We first characterize the conditional error and conditional regret of the deferral loss as follows.

**Lemma 5.1.** *Assume that $\mathcal{H}$ is symmetric and complete. Then, for any $h \in \mathcal{H}$ and input $x \in \mathcal{X}$, the conditional error and conditional regret of the deferral loss are given by:*

$$\mathcal{C}_{\mathsf{L}_{\mathrm{def}}}(h, x) = 1 - p_{h(x)}, \quad \Delta\mathcal{C}_{\mathsf{L}_{\mathrm{def}}, \mathcal{H}}^*(h, x) = \max\left\{p_{y_{\max}}, \max_{j \in [n_e]} p_{n+j}\right\} - p_{h(x)}.$$

*Proof.* By definition,

$$\mathcal{C}_{\mathsf{L}_{\mathrm{def}}}(h, x) = \sum_{y \in \mathcal{Y}} p(y|x) \mathsf{L}_{\mathrm{def}}(h, x, y)$$

$$= \sum_{y \in \mathcal{Y}} p(y|x) 1_{h(x) \neq y} 1_{h(x) \in [n]} + \sum_{j=1}^{n_e} \sum_{y \in \mathcal{Y}} p(y|x) c_j(x, y) 1_{h(x) = n+j}$$

$$= (1 - p(h(x)|x) 1_{h(x) \in [n]} + \sum_{j=1}^{n_e} (1 - p(n+j|x)) 1_{h(x) = n+j}$$

$$= 1 - p_{h(x)}.$$

Therefore, the best-in-class conditional error and conditional regret can be expressed as follows:

$$\mathcal{C}^*_{\mathsf{L}_{\mathrm{def}}}(\mathcal{H}, x) = 1 - \max\left\{p_{y_{\max}}, \max_{j \in [n_e]} p_{n+j}\right\}, \quad \Delta \mathcal{C}^*_{\mathsf{L}_{\mathrm{def}}, \mathcal{H}}(h, x) = \max\left\{p_{y_{\max}}, \max_{j \in [n_e]} p_{n+j}\right\} - p_{\mathsf{h}(x)}.$$

$\square$

For convenience, in the following sections, we will omit the dependency on $x$ in the notation: $h_y = h(x, y)$ for any $y \in \overline{\mathcal{Y}}$, and $p_y = p(y|x)$ for any $y \in \mathcal{Y}$. We let $h_{\max} = \mathrm{argmax}_{y \in \mathcal{Y}} h_y$ and $q_{j,y} = p(y|x) c_j(x, y)$ for any $y \in \mathcal{Y}$. We also let $y^* = \mathrm{argmax}_{j \in [n_e]} p_{n+j}$ and $h^* = \mathrm{argmax}_{j \in [n_e]} h_{n+j}$.

### B.3.2. PROOF OF THEOREM 3.4

**Theorem 3.4.** *Assume that $\mathcal{H}$ is symmetric and complete. Then, for all $h \in \mathcal{H}$ and any distribution, the following $\mathcal{H}$-consistency bound holds:*

$$\mathcal{E}_{\mathsf{L}_{\mathrm{def}}}(h) - \mathcal{E}_{\mathsf{L}_{\mathrm{def}}}(\mathcal{H}) + \mathcal{M}_{\mathsf{L}_{\mathrm{def}}}(\mathcal{H}) \le (n + n_e)(\mathcal{E}_{\mathsf{L}_{\mathrm{mae}}}(h) - \mathcal{E}_{\mathsf{L}_{\mathrm{mae}}}(\mathcal{H}) + \mathcal{M}_{\mathsf{L}_{\mathrm{mae}}}(\mathcal{H})).$$

*Proof.* The conditional error of the surrogate loss can be expressed as follows:

$$
\begin{aligned}
&\mathcal{C}_{\mathsf{L}_{\mathrm{mae}}}(h, x) \\
&= \sum_{y \in \mathcal{Y}} p(y|x) \mathsf{L}_{\mathrm{mae}}(h, x, y) \\
&= \sum_{j=1}^{n_e} \left[ \sum_{y \in \mathcal{Y}} p(y|x) c_j(x, y) \left(1 - \frac{e^{h(x,y)}}{\sum_{y' \in \overline{\mathcal{Y}}} e^{h(x,y')}}\right) + \sum_{y \in \mathcal{Y}} p(y|x)(1 - c_j(x, y)) \left(1 - \frac{e^{h(x,y)} + e^{h(x,n+j)}}{\sum_{y' \in \overline{\mathcal{Y}}} e^{h(x,y')}}\right) \right] \\
&\quad + (1 - n_e) \sum_{y \in \mathcal{Y}} p(y|x) \left(1 - \frac{e^{h(x,y)}}{\sum_{y' \in \overline{\mathcal{Y}}} e^{h(x,y')}}\right) \\
&= \sum_{j=1}^{n_e} \left[ \sum_{y \in \mathcal{Y}} q_{j,y} \left(1 - \frac{e^{h_y}}{\sum_{y' \in \overline{\mathcal{Y}}} e^{h_{y'}}}\right) + \sum_{y \in \mathcal{Y}} (p_y - q_{j,y}) \left(1 - \frac{e^{h_y} + e^{h_{n+j}}}{\sum_{y' \in \overline{\mathcal{Y}}} e^{h_{y'}}}\right) \right] + (1 - n_e) \sum_{y \in \mathcal{Y}} p_y \left(1 - \frac{e^{h_y}}{\sum_{y' \in \overline{\mathcal{Y}}} e^{h_{y'}}}\right).
\end{aligned}
$$

By Lemma 5.1, we can write the conditional regret of the deferral loss as

$$\Delta \mathcal{C}_{\mathsf{L}_{\mathrm{def}}, \mathcal{H}}(h, x) = \max\left\{p_{y_{\max}}, \max_{j \in [n_e]} p_{n+j}\right\} - p_{\mathsf{h}(x)}.$$

Next, we show that the surrogate conditional regret can be lower bounded by the target one:

$$\Delta \mathcal{C}_{\mathsf{L}_{\mathrm{mae}}, \mathcal{H}}(h, x) = \mathcal{C}_{\mathsf{L}_{\mathrm{mae}}}(h) - \mathcal{C}^*_{\mathsf{L}_{\mathrm{mae}}}(\mathcal{H}) \ge \frac{1}{n + n_e}(\Delta \mathcal{C}_{\mathsf{L}_{\mathrm{def}}, \mathcal{H}}(h, x)). \tag{10}$$

We first prove that for any hypothesis $h$ and $x \in \mathcal{X}$, if $y_{\max} \ne h_{\max}$, then the conditional error of $h$ can be lower bounded by

that of $\overline{h}$, which satisfies that $\overline{h}(x,y) = \begin{cases} h_{h_{\max}} & y = y_{\max} \\ h_{y_{\max}} & y = h_{\max} \\ h_y & \text{otherwise.} \end{cases}$ . Indeed,

$$
\begin{aligned}
\mathcal{C}_{\mathsf{L}_{\mathrm{mae}}}(h) - \mathcal{C}_{\mathsf{L}_{\mathrm{mae}}}(\overline{h}) = \sum_{j=1}^{n_e} \Bigg[ & q_{j,y_{\max}}\left(1 - \frac{e^{h_{y_{\max}}}}{\sum_{y'\in\overline{y}} e^{h_{y'}}}\right) + (p_{y_{\max}} - q_{j,y_{\max}})\left(1 - \frac{e^{h_{y_{\max}}} + e^{h_{n+j}}}{\sum_{y'\in\overline{y}} e^{h_{y'}}}\right) \\
& + q_{j,h_{\max}}\left(1 - \frac{e^{h_{h_{\max}}}}{\sum_{y'\in\overline{y}} e^{h_{y'}}}\right) + (p_{h_{\max}} - q_{j,h_{\max}})\left(1 - \frac{e^{h_{h_{\max}}} + e^{h_{n+j}}}{\sum_{y'\in\overline{y}} e^{h_{y'}}}\right) \\
& - q_{j,y_{\max}}\left(1 - \frac{e^{h_{h_{\max}}}}{\sum_{y'\in\overline{y}} e^{h_{y'}}}\right) - (p_{y_{\max}} - q_{j,y_{\max}})\left(1 - \frac{e^{h_{h_{\max}}} + e^{h_{n+j}}}{\sum_{y'\in\overline{y}} e^{h_{y'}}}\right) \\
& - q_{j,h_{\max}}\left(1 - \frac{e^{h_{y_{\max}}}}{\sum_{y'\in\overline{y}} e^{h_{y'}}}\right) - (p_{h_{\max}} - q_{j,h_{\max}})\left(1 - \frac{e^{h_{y_{\max}}} + e^{h_{n+j}}}{\sum_{y'\in\overline{y}} e^{h_{y'}}}\right)\Bigg] \\
& + (1-n_e)p_{y_{\max}}\left(1 - \frac{e^{y_{\max}}}{\sum_{y'\in\overline{y}} e^{h_{y'}}}\right) + (1-n_e)p_{h_{\max}}\left(1 - \frac{e^{h_{\max}}}{\sum_{y'\in\overline{y}} e^{h_{y'}}}\right) \\
& - (1-n_e)p_{y_{\max}}\left(1 - \frac{e^{h_{\max}}}{\sum_{y'\in\overline{y}} e^{h_{y'}}}\right) - (1-n_e)p_{h_{\max}}\left(1 - \frac{e^{y_{\max}}}{\sum_{y'\in\overline{y}} e^{h_{y'}}}\right) \\
= & \frac{1}{\sum_{y'\in\overline{y}} e^{h_{y'}}}(p_{y_{\max}} - p_{h_{\max}})\left(e^{h_{h_{\max}}} - e^{h_{y_{\max}}}\right) \geq 0.
\end{aligned}
$$

We then prove that for any hypothesis $h$ and $x \in \mathcal{X}$, if $y^* \neq h^*$, then the conditional error of $h$ can be lower bounded by that of $\widetilde{h}$, which satisfies that $\widetilde{h}(x,y) = \begin{cases} h_{n+h^*} & y = n + y^* \\ h_{n+y^*} & y = n + h^* \\ h_y & \text{otherwise.} \end{cases}$ . Indeed,

$$
\begin{aligned}
\mathcal{C}_{\mathsf{L}_{\mathrm{mae}}}(h) - \mathcal{C}_{\mathsf{L}_{\mathrm{mae}}}(\widetilde{h}) = & \; p_{n+y^*}\left(1 - \frac{e^{h_{n+y^*}}}{\sum_{y'\in\overline{y}} e^{h_{y'}}}\right) + p_{n+h^*}\left(1 - \frac{e^{h_{n+h^*}}}{\sum_{y'\in\overline{y}} e^{h_{y'}}}\right) \\
& - p_{n+y^*}\left(1 - \frac{e^{h_{n+h^*}}}{\sum_{y'\in\overline{y}} e^{h_{y'}}}\right) - p_{n+h^*}\left(1 - \frac{e^{h_{n+y^*}}}{\sum_{y'\in\overline{y}} e^{h_{y'}}}\right) \\
= & \; \frac{1}{\sum_{y'\in\overline{y}} e^{h_{y'}}}(p_{n+y^*} - p_{n+h^*})\left(e^{h_{n+y^*}} - e^{h_{n+y^*}}\right) \geq 0.
\end{aligned}
$$

Therefore, we only need to lower bound the conditional regret of hypothesis $h$ satisfying both $y_{\max} = h_{\max}$ and $y^* = h^*$. Note that if $(p_{y_{\max}} - p_{n+y^*})(h_{y_{\max}} - h_{n+y^*}) > 0$, then, $\Delta\mathcal{C}_{\mathsf{L}_{\mathrm{def}},\mathcal{H}}(h,x) = 0$. Next, we will analyze case by case.

1. **Case I**: If $p_{y_{\max}} - p_{n+y^*} \geq 0$ and $h_{y_{\max}} - h_{n+y^*} \leq 0$: we define a new hypothesis $h_\mu$ such that $h_\mu(x,y) = \begin{cases} \log(e^{h_{n+y^*}} + \mu) & y = y_{\max} \\ \log(e^{h_{y_{\max}}} - \mu) & y = n + y^* \\ h(x,y) & \text{otherwise.} \end{cases}$, where $e^{h_{y_{\max}}} \geq \mu \geq 0$. Then, we can lower bound the conditional regret of $\mathsf{L}_{\mathrm{mae}}$ by

using $\Delta \mathcal{C}_{\mathsf{L}_{\mathrm{mae}},\mathcal{H}}(h,x) \geq \mathcal{C}_{\mathsf{L}_{\mathrm{mae}}}(h) - \mathcal{C}^*_{\mathsf{L}_{\mathrm{mae}}}(h_\mu)$ for any $e^{h_{y_{\max}}} \geq \mu \geq 0$:

$$
\begin{aligned}
&\Delta \mathcal{C}_{\mathsf{L}_{\mathrm{mae}},\mathcal{H}}(h,x) \\
&\geq \sup_{e^{h_{y_{\max}}} \geq \mu \geq 0} \left( \mathcal{C}_{\mathsf{L}_{\mathrm{mae}}}(h) - \mathcal{C}^*_{\mathsf{L}_{\mathrm{mae}}}(h_\mu) \right) \\
&\geq \sup_{e^{h_{y_{\max}}} \geq \mu \geq 0} \left( \sum_{j \neq y^*}^{n_e} \left[ q_{j,y_{\max}} \left( 1 - \frac{e^{h_{y_{\max}}}}{\sum_{y' \in \overline{y}} e^{h_{y'}}} \right) + (p_{y_{\max}} - q_{j,y_{\max}}) \left( 1 - \frac{e^{h_{y_{\max}}} + e^{h_{n+j}}}{\sum_{y' \in \overline{y}} e^{h_{y'}}} \right) \right. \right. \\
&\qquad \left. - q_{j,y_{\max}} \left( 1 - \frac{e^{h_{n+y^*}} + \mu}{\sum_{y' \in \overline{y}} e^{h_{y'}}} \right) - (p_{y_{\max}} - q_{j,y_{\max}}) \left( 1 - \frac{e^{h_{n+y^*}} + \mu + e^{h_{n+j}}}{\sum_{y' \in \overline{y}} e^{h_{y'}}} \right) \right] \\
&\qquad + q_{y^*,y_{\max}} \left( 1 - \frac{e^{h_{y_{\max}}}}{\sum_{y' \in \overline{y}} e^{h_{y'}}} \right) + (p_{y_{\max}} - q_{y^*,y_{\max}}) \left( 1 - \frac{e^{h_{y_{\max}}} + e^{h_{n+y^*}}}{\sum_{y' \in \overline{y}} e^{h_{y'}}} \right) \\
&\qquad - q_{y^*,y_{\max}} \left( 1 - \frac{e^{h_{n+y^*}} + \mu}{\sum_{y' \in \overline{y}} e^{h_{y'}}} \right) - (p_{y_{\max}} - q_{y^*,y_{\max}}) \left( 1 - \frac{e^{h_{n+y^*}} + e^{h_{y_{\max}}}}{\sum_{y' \in \overline{y}} e^{h_{y'}}} \right) \\
&\qquad + \sum_{y \neq y_{\max}} (p_y - q_{y^*,y}) \left( 1 - \frac{e^{h_y} + e^{h_{n+y^*}}}{\sum_{y' \in \overline{y}} e^{h_{y'}}} \right) - \sum_{y \neq y_{\max}} (p_y - q_{y^*,y}) \left( 1 - \frac{e^{h_y} + e^{h_{y_{\max}}} - \mu}{\sum_{y' \in \overline{y}} e^{h_{y'}}} \right) \\
&\qquad \left. + (1 - n_e) \frac{p_{y_{\max}}}{\sum_{y' \in \overline{y}} e^{h_{y'}}} \left( e^{h_{n+y^*}} + \mu - e^{h_{y_{\max}}} \right) \right) \\
&= \frac{1}{\sum_{y' \in \overline{y}} e^{h_{y'}}} \sup_{e^{h_{y_{\max}}} \geq \mu \geq 0} \left( \left[ q_{y^*,y_{\max}} - \sum_{y \neq y_{\max}} (p_y - q_{y^*,y}) \right] \left( e^{h_{n+y^*}} + \mu - e^{h_{y_{\max}}} \right) \right) \\
&= (p_{y_{\max}} - p_{n+y^*}) \frac{e^{h_{n+y^*}}}{\sum_{y' \in \overline{y}} e^{h_{y'}}} \qquad\qquad (\mu = e^{h_{y_{\max}}} \text{ achieves the maximum}) \\
&\geq \frac{1}{n + n_e} (p_{y_{\max}} - p_{n+y^*}) \qquad\qquad \text{(by the assumption } h_{n+y^*} = h_{n+h^*} \geq h_{y_{\max}} = h_{h_{\max}}) \\
&= \frac{1}{n + n_e} (\Delta \mathcal{C}_{\mathsf{L}_{\mathrm{def}},\mathcal{H}}(h,x)) \qquad\qquad \text{(by the assumption } p_{y_{\max}} - p_{n+y^*} \geq 0 \text{ and } h_{y_{\max}} - h_{n+y^*} \leq 0)
\end{aligned}
$$

2. **Case II**: If $p_{y_{\max}} - p_{n+y^*} \leq 0$ and $h_{y_{\max}} - h_{n+y^*} \geq 0$: we define a new hypothesis $h_\mu$ such that $h_\mu(x,y) = $
$$
\begin{cases}
\log(e^{h_{n+y^*}} - \mu) & y = y_{\max} \\
\log(e^{h_{y_{\max}}} + \mu) & y = n + y^*, \text{ where } e^{h_{n+y^*}} \geq \mu \geq 0. \text{ Then, we can lower bound the conditional regret of } \mathsf{L}_{\mathrm{mae}} \\
h(x,y) & \text{otherwise.}
\end{cases}
$$

by using $\Delta\mathcal{C}_{\mathsf{L}_{\mathrm{mae}},\mathcal{H}}(h,x) \geq \mathcal{C}_{\mathsf{L}_{\mathrm{mae}}}(h) - \mathcal{C}^*_{\mathsf{L}_{\mathrm{mae}}}(h_\mu)$ for any $e^{h_{n+y^*}} \geq \mu \geq 0$:

$$
\begin{aligned}
&\Delta\mathcal{C}_{\mathsf{L}_{\mathrm{mae}},\mathcal{H}}(h,x) \\
&\geq \sup_{e^{h_{n+y^*}} \geq \mu \geq 0} \left( \mathcal{C}_{\mathsf{L}_{\mathrm{mae}}}(h) - \mathcal{C}^*_{\mathsf{L}_{\mathrm{mae}}}(h_\mu) \right) \\
&\geq \sup_{e^{h_{n+y^*}} \geq \mu \geq 0} \left( \sum_{j\neq y^*}^{n_e} \left[ q_{j,y_{\max}}\left(1 - \frac{e^{h_{y_{\max}}}}{\sum_{y'\in\overline{\mathcal{y}}} e^{h_{y'}}}\right) + (p_{y_{\max}} - q_{j,y_{\max}})\left(1 - \frac{e^{h_{y_{\max}}} + e^{h_{n+j}}}{\sum_{y'\in\overline{\mathcal{y}}} e^{h_{y'}}}\right) \right.\right. \\
&\qquad\qquad \left. - q_{j,y_{\max}}\left(1 - \frac{e^{h_{n+y^*}} - \mu}{\sum_{y'\in\overline{\mathcal{y}}} e^{h_{y'}}}\right) - (p_{y_{\max}} - q_{j,y_{\max}})\left(1 - \frac{e^{h_{n+y^*}} - \mu + e^{h_{n+j}}}{\sum_{y'\in\overline{\mathcal{y}}} e^{h_{y'}}}\right) \right] \\
&\qquad\qquad + q_{y^*,y_{\max}}\left(1 - \frac{e^{h_{y_{\max}}}}{\sum_{y'\in\overline{\mathcal{y}}} e^{h_{y'}}}\right) + (p_{y_{\max}} - q_{y^*,y_{\max}})\left(1 - \frac{e^{h_{y_{\max}}} + e^{h_{n+y^*}}}{\sum_{y'\in\overline{\mathcal{y}}} e^{h_{y'}}}\right) \\
&\qquad\qquad - q_{y^*,y_{\max}}\left(1 - \frac{e^{h_{n+y^*}} - \mu}{\sum_{y'\in\overline{\mathcal{y}}} e^{h_{y'}}}\right) - (p_{y_{\max}} - q_{y^*,y_{\max}})\left(1 - \frac{e^{h_{n+y^*}} + e^{h_{y_{\max}}}}{\sum_{y'\in\overline{\mathcal{y}}} e^{h_{y'}}}\right) \\
&\qquad\qquad + \sum_{y\neq y_{\max}} (p_y - q_{y^*,y})\left(1 - \frac{e^{h_y} + e^{h_{n+y^*}}}{\sum_{y'\in\overline{\mathcal{y}}} e^{h_{y'}}}\right) - \sum_{y\neq y_{\max}} (p_y - q_{y^*,y})\left(1 - \frac{e^{h_y} + e^{h_{y_{\max}}} + \mu}{\sum_{y'\in\overline{\mathcal{y}}} e^{h_{y'}}}\right) \\
&\qquad\qquad \left. + (1 - n_e)\frac{p_{y_{\max}}}{\sum_{y'\in\overline{\mathcal{y}}} e^{h_{y'}}}\left(e^{h_{n+y^*}} - \mu - e^{h_{y_{\max}}}\right) \right) \\
&= \frac{1}{\sum_{y'\in\overline{\mathcal{y}}} e^{h_{y'}}} \sup_{e^{h_{y_{\max}}} \geq \mu \geq 0} \left( \left[ \sum_{y\neq y_{\max}} (p_y - q_{y^*,y}) - q_{y^*,y_{\max}} \right]\left(e^{h_{y_{\max}}} + \mu - e^{h_{n+y^*}}\right) \right) \\
&= (p_{n+y^*} - p_{y_{\max}})\frac{e^{h_{y_{\max}}}}{\sum_{y'\in\overline{\mathcal{y}}} e^{h_{y'}}} \qquad\qquad (\mu = e^{h_{n+y^*}} \text{ achieves the maximum}) \\
&\geq \frac{1}{n + n_e}(p_{n+y^*} - p_{y_{\max}}) \qquad\qquad (\text{by the assumption } h_{y_{\max}} = h_{h_{\max}} \geq h_{n+y^*} = h_{n+h^*}) \\
&= \frac{1}{n + n_e}\left(\Delta\mathcal{C}_{\mathsf{L}_{\mathrm{def}},\mathcal{H}}(h,x)\right) \qquad (\text{by the assumption } p_{y_{\max}} - p_{n+y^*} \leq 0 \text{ and } h_{y_{\max}} - h_{n+y^*} \geq 0)
\end{aligned}
$$

This proves the inequality (10). By taking the expectation on both sides of (10), we complete the proof. $\qquad\square$

## C. Two-stage multiple-expert deferral: proofs.

### C.1. Proof of Theorem 4.1

**Theorem 4.1.** *Assume that $\mathcal{R}$ is closed under scaling and that $\Phi$ satisfies $\lim_{t\to+\infty}\Phi(t) = 0$ and $\Phi(t) \geq 1_{t\leq 0}$. Then, the surrogate loss $\mathsf{L}_\Phi$ is realizable $\mathcal{R}$-consistent with respect to $\mathsf{L}_{\mathrm{tdef}}$.*

*Proof.* Let $r^*$ be a best-in-class predictor such that $\mathcal{E}_{\mathsf{L}_{\mathrm{tdef}}}(r^*) = 0$. Note that

$$
\begin{aligned}
\mathcal{E}^*_{\mathsf{L}_\Phi}(\mathcal{R}) &\leq \lim_{\alpha\to+\infty} \mathcal{E}_{\mathsf{L}_\Phi}(\alpha r^*) \\
&= \lim_{\alpha\to+\infty} \mathbb{E}\left[\mathsf{L}_\Phi(\alpha r^*, x, y) \mid r^*(x) = 1\right]\mathbb{P}(r^*(x) = 1) \\
&\qquad + \mathbb{E}\left[\mathsf{L}_\Phi(\alpha r^*, x, y) \mid r^*(x) = 2\right]\mathbb{P}(r^*(x) = 2).
\end{aligned}
$$

If $r^*(x) = 1$, then, we must have $c_1(x, y) = 0$ for any $(x, y) \in \mathcal{X} \times \mathcal{Y}$. In this case,

$$\lim_{\alpha \to +\infty} \mathbb{E}[\mathsf{L}_\Phi(\alpha r^*, x, y) \mid r^*(x) = 1]$$
$$\leq \mathbb{E}\left[c_2(x, y) \lim_{\alpha \to +\infty} \Phi(\alpha(r(x, 1) - r(x, 2)))\right]$$
$$= \mathbb{E}[0]$$
$$= 0.$$

If $r^*(x) = 2$, then, we must have $c_2(x, y) = 0$ for any $(x, y) \in \mathcal{X} \times \mathcal{Y}$. In this case,

$$\lim_{\alpha \to +\infty} \mathbb{E}[\mathsf{L}_\Phi(\alpha r^*, x, y) \mid r^*(x) = 2]$$
$$\leq \mathbb{E}\left[c_1(x, y) \lim_{\alpha \to +\infty} \Phi(\alpha(r(x, 2) - r(x, 1)))\right]$$
$$= \mathbb{E}[0]$$
$$= 0.$$

Therefore, we have $\mathcal{E}^*_{\mathsf{L}_\Phi}(\mathcal{R}) = 0$. $\qquad\qquad\qquad\qquad\qquad\qquad\qquad\qquad\qquad\qquad\qquad\qquad\qquad\qquad$ $\square$

### C.2. Proof of Theorem 4.2

**Theorem 4.2.** *Assume that the following $\mathcal{R}$-consistency bound holds in binary classification:*

$$\mathcal{E}_{\ell_{0-1}}(h) - \mathcal{E}_{\ell_{0-1}}(\mathcal{R}) + \mathcal{M}_{\ell_{0-1}}(\mathcal{R}) \leq \Gamma(\mathcal{E}_\Phi(h) - \mathcal{E}_\Phi(\mathcal{R}) + \mathcal{M}_\Phi(\mathcal{R})).$$

*Then, the following $\mathcal{R}$-consistency bound holds in two-stage, two-expert deferral:*

$$\mathcal{E}_{\mathsf{L}_{\mathrm{tdef}}}(r) - \mathcal{E}_{\mathsf{L}_{\mathrm{tdef}}}(\mathcal{R}) + \mathcal{M}_{\mathsf{L}_{\mathrm{tdef}}}(\mathcal{R}) \leq (\overline{c}_1 + \overline{c}_2)\Gamma\left(\frac{\mathcal{E}_{\mathsf{L}_\Phi}(r) - \mathcal{E}_{\mathsf{L}_\Phi}(\mathcal{R}) + \mathcal{M}_{\mathsf{L}_\Phi}(\mathcal{R})}{\underline{c}_1 + \underline{c}_2}\right),$$

*where constant factors $(\underline{c}_1 + \underline{c}_2)$ and $(\overline{c}_1 + \overline{c}_2)$ can be removed when $\Gamma$ is linear.*

*Proof.* Let $\eta(x) = \mathbb{P}(Y = 1 \mid X = x)$ be the conditional probability of $Y = 1$ given $X = x$. The conditional error of $\mathsf{L}_{\mathsf{L}_{\mathrm{tdef}}}$ can be expressed as follows:

$$\mathcal{C}_{\mathsf{L}_{\mathsf{L}_{\mathrm{tdef}}}}(r, x) = \eta(x)\left[c_1(x, 1)1_{r(x)=1} + c_2(x, 1)1_{r(x)=2}\right]$$
$$+ (1 - \eta(x))\left[c_1(x, 2)1_{r(x)=1} + c_2(x, 2)1_{r(x)=2}\right]$$
$$= \left[\eta(x)c_1(x, 1) + (1 - \eta(x))c_1(x, 2)\right]1_{r(x)=1}$$
$$+ \left[\eta(x)c_2(x, 1) + (1 - \eta(x))c_2(x, 2)\right]1_{r(x)=2}$$

The conditional error of $\mathsf{L}_\Phi$ can be expressed as follows:

$$\mathcal{C}_{\mathsf{L}_\Phi}(r, x) = \eta(x)\left[c_1(x, 1)\Phi(r(x, 2) - r(x, 1)) + c_2(x, 1)\Phi(r(x, 1) - r(x, 2))\right]$$
$$+ (1 - \eta(x))\left[c_1(x, 2)\Phi(r(x, 2) - r(x, 1)) + c_2(x, 2)\Phi(r(x, 1) - r(x, 2))\right]$$
$$= \left[\eta(x)c_1(x, 1) + (1 - \eta(x))c_1(x, 2)\right]\Phi(r(x, 2) - r(x, 1))$$
$$+ \left[\eta(x)c_2(x, 1) + (1 - \eta(x))c_2(x, 2)\right]\Phi(r(x, 1) - r(x, 2))$$

Consider a new distribution which satisfies that

$$\widetilde{\eta}(x) = \frac{\eta(x)c_1(x, 1) + (1 - \eta(x))c_1(x, 2)}{[\eta(x)c_1(x, 1) + (1 - \eta(x))c_1(x, 2)] + [\eta(x)c_2(x, 1) + (1 - \eta(x))c_2(x, 2)]}.$$

Then, under the assumption, we have

$$\widetilde{\eta}(x)1_{\mathsf{r}(x)=1} + (1 - \widetilde{\eta}(x))1_{\mathsf{r}(x)=2} - \inf_{r \in \mathcal{R}}\left[\widetilde{\eta}(x)1_{\mathsf{r}(x)=1} + (1 - \widetilde{\eta}(x))1_{\mathsf{r}(x)=2}\right]$$

$$\leq \Gamma\Bigg(\widetilde{\eta}(x)\Phi(r(x,2) - r(x,1)) + (1 - \widetilde{\eta}(x))\Phi(r(x,1) - r(x,2))$$

$$- \inf_{r \in \mathcal{R}}\left[\widetilde{\eta}(x)\Phi(r(x,2) - r(x,1)) + (1 - \widetilde{\eta}(x))\Phi(r(x,1) - r(x,2))\right]\Bigg).$$

Using the fact that $[\eta(x)c_1(x,1) + (1 - \eta(x))c_1(x,2)] + [\eta(x)c_2(x,1) + (1 - \eta(x))c_2(x,2)] \in [\underline{c}_1 + \underline{c}_2, \overline{c}_1 + \overline{c}_2]$, we obtain

$$\mathcal{C}_{\mathsf{L}_{\mathrm{tdef}}}(r,x) - \inf_{r \in \mathcal{R}}\mathcal{C}_{\mathsf{L}_{\mathrm{tdef}}}(r,x) \leq (\overline{c}_1 + \overline{c}_2)\Gamma\left(\frac{\mathcal{C}_{\mathsf{L}_\Phi}(r,x) - \inf_{r \in \mathcal{R}}\mathcal{C}_{\mathsf{L}_\Phi}(r,x)}{\underline{c}_1 + \underline{c}_2}\right),$$

where constant factors $(\underline{c}_1 + \underline{c}_2)$ and $(\overline{c}_1 + \overline{c}_2)$ can be removed when $\Gamma$ is linear. By taking the expectation on both sides and applying Jensen's inequality, we conclude the proof. $\qquad\square$

## C.3. Proof of Theorem 4.3

**Theorem 4.3.** *Assume that $\mathcal{R}$ is closed under scaling and that $c_j(x,y) = 1_{\mathsf{g}_j(x)\neq y}, j \in [n_e]$. Further, assume that for any $(x,y)$, there is at most one expert $j^* \in [n_e]$ for which $c_{j^*}(x,y) = 0$ and that $\Psi$ satisfies $\lim_{u \to 1^-} \Psi(u) = 0$. Then, the surrogate loss $\mathsf{L}_\Psi$ is realizable $\mathcal{R}$-consistent with respect to $\mathsf{L}_{\mathrm{tdef}}$.*

*Proof.* Let $r^*$ be a best-in-class predictor such that $\mathcal{E}_{\mathsf{L}_{\mathrm{tdef}}}(r^*) = 0$. Since $\mathcal{R}$ is closed under scaling, for any $\alpha$, $\alpha r^*$ is in $\mathcal{R}$, and we have $\mathcal{E}^*_{\mathsf{L}_\Psi}(\mathcal{R}) \leq \mathcal{E}_{\mathsf{L}_\Psi}(\alpha r^*)$. This implies the following inequality:

$$\mathcal{E}^*_{\mathsf{L}_\Psi}(\mathcal{R}) \leq \lim_{\alpha \to +\infty}\mathcal{E}_{\mathsf{L}_\Psi}(\alpha r^*)$$

$$= \sum_{j=1}^{n_e}\lim_{\alpha \to +\infty}\mathbb{E}[\mathsf{L}_\Psi(\alpha r^*, x, y) \mid \mathsf{r}^*(x) = i]\mathbb{P}(\mathsf{r}^*(x) = i).$$

If $\mathsf{r}^*(x) = i$, then, we must have $c_i(x,y) = 0$ and $c_k(x,y) = 1$, for all $k \neq i$ and all $(x,y) \in \mathcal{X} \times \mathcal{Y}$. Thus, we can write

$$\lim_{\alpha \to +\infty}\mathbb{E}[\mathsf{L}_\Psi(\alpha r^*, x, y) \mid \mathsf{r}^*(x) = i]$$

$$\leq \mathbb{E}\left[\sum_{j=1}^{n_e}\left(\sum_{j' \neq j}c_{j'}(x,y) - n_e + 2\right)\lim_{\alpha \to +\infty}\Psi\left(\frac{e^{\alpha r^*(x,j)}}{\sum_{j' \in [n_e]}e^{\alpha r^*(x,j')}}\right) \Big| \mathsf{r}^*(x) = i\right] \qquad \text{(By (7))}$$

$$= \mathbb{E}\left[\lim_{\alpha \to +\infty}\Psi\left(\frac{e^{\alpha r^*(x,i)}}{\sum_{j' \in [n_e]}e^{\alpha r^*(x,j')}}\right)\right] \qquad\qquad (c_i(x,y) = 0 \text{ and } c_k(x,y) = 1, \forall k \neq i)$$

$$= \mathbb{E}[0] \qquad\qquad (\mathsf{r}^*(x) = i \implies r^*(x,i) > r^*(x,j'), \forall j' \neq i, \lim_{u \to 1^-}\Psi(u) = 0)$$

$$= 0.$$

Therefore, we have $\mathcal{E}^*_{\mathsf{L}_\Psi}(\mathcal{R}) = 0$. $\qquad\square$

## C.4. Proof of $\mathcal{H}$-consistency bounds

### C.4.1. SIMPLIFIED NOTATION AND LEMMA C.1

Let $p(y|x) = \mathbb{P}(Y = y \mid X = x)$ be the conditional probability of $Y = y$ given $X = x$. We first characterize the best-in class conditional error and the conditional regret of the two-stage deferral loss function $\mathsf{L}_{\mathrm{tdef}}$, which will be used in the analysis of $\mathcal{H}$-consistency bounds.

**Lemma C.1.** *Assume that $\mathcal{R}$ is symmetric and complete. Then, for any $r \in \mathcal{K}$ and $x \in \mathcal{X}$, the best-in class conditional error and the conditional regret of the two-stage deferral loss function are given by:*

$$\mathcal{C}^*_{\mathsf{L}_{\mathrm{tdef}}}(\mathcal{R},x) = \min_{j \in [n_e]}\sum_{y \in \mathcal{Y}}p(y|x)c_j(x,y), \quad \Delta\mathcal{C}_{\mathsf{L}_{\mathrm{tdef}},\mathcal{R}}(r,x) = \sum_{y \in \mathcal{Y}}p(y|x)c_{\mathsf{r}(x)}(x,y) - \min_{j \in [n_e]}\sum_{y \in \mathcal{Y}}p(y|x)c_j(x,y).$$

*Proof.* By definition, for any $r \in \mathcal{R}$ and $x \in \mathcal{X}$, the conditional error of the two-stage deferral loss function can be written as

$$\mathcal{C}_{\mathsf{L}_{\mathrm{tdef}}}(r, x) = \sum_{y \in \mathcal{Y}} p(y|x) c_{\mathsf{r}(x)}(x, y).$$

Since $\mathcal{R}$ is symmetric and complete, we have

$$\mathcal{C}^*_{\mathsf{L}_{\mathrm{tdef}}}(\mathcal{R}, x) = \inf_{r \in \mathcal{R}} \sum_{y \in \mathcal{Y}} p(y|x) c_{\mathsf{r}(x)}(x, y) = \min_{j \in [n_e]} \sum_{y \in \mathcal{Y}} p(y|x) c_j(x, y).$$

Furthermore, the conditional regret can be expressed as

$$\Delta \mathcal{C}_{\mathsf{L}_{\mathrm{tdef}}, \mathcal{R}}(r, x) = \mathcal{C}_{\mathsf{L}_{\mathrm{tdef}}}(r, x) - \mathcal{C}^*_{\mathsf{L}_{\mathrm{tdef}}}(\mathcal{R}, x) = \sum_{y \in \mathcal{Y}} p(y|x) c_{\mathsf{r}(x)}(x, y) - \min_{j \in [n_e]} \sum_{y \in \mathcal{Y}} p(y|x) c_j(x, y),$$

which completes the proof. $\qquad \square$

### C.4.2. PROOF OF THEOREM 4.4

For convenience, we let $\overline{C} = (n_e - 1) \max_{j \in [n_e]} \overline{c}_j - n_e + 2$, $\overline{c}_j(x, y) = \sum_{j' \neq j} c_{j'}(x, y) - n_e + 2 \in [0, \overline{C}]$, $\overline{q}(x, j) = \sum_{y \in \mathcal{Y}} p(y|x) \overline{c}_j(x, y) \in [0, \overline{C}]$ and $\mathcal{S}(x, j) = \frac{e^{r(x,j)}}{\sum_{j' \in [n_e]} e^{r(x,j')}}$. We also let $j_{\min}(x) = \operatorname{argmin}_{j \in [n_e]} \sum_{y \in \mathcal{Y}} p(y|x) c_j(x, y)$.

**Theorem 4.4.** *Assume that $\mathcal{R}$ is symmetric and complete and that the inequality $\sum_{j' \neq j} c_{j'}(x, y) \geq n_e - 2$ holds for all $j \in [n_e]$ and $(x, y) \in \mathcal{X} \times \mathcal{Y}$. Then, the following $\mathcal{R}$-consistency bound holds:*

$$\mathcal{E}_{\mathsf{L}_{\mathrm{tdef}}}(r) - \mathcal{E}^*_{\mathsf{L}_{\mathrm{tdef}}}(\mathcal{R}) + \mathcal{M}_{\mathsf{L}_{\mathrm{tdef}}}(\mathcal{R}) \leq \Gamma\Big(\mathcal{E}_{\mathsf{L}_{\Psi_q}}(r) - \mathcal{E}^*_{\mathsf{L}_{\Psi_q}}(\mathcal{R}) + \mathcal{M}_{\mathsf{L}_{\Psi_q}}(\mathcal{R})\Big),$$

*where $\overline{C} = (n_e - 1) \max_{j \in [n_e]} \overline{c}_j - n_e + 2$, $\Gamma(t) = 2(\overline{C})^{\frac{1}{2}} \sqrt{t}$ when $q = 0$, $\Gamma(t) = 2(n_e{}^q)^{\frac{1}{2}} (\overline{C})^{\frac{1}{2}} \sqrt{t}$ when $q \in (0, 1)$, and $\Gamma(t) = n_e t$ when $q = 1$.*

*Proof.* **Case I:** $q = 0$. In this case, for the surrogate loss $\mathsf{L}_{\Psi_q}$, the conditional error can be written as follows:

$$\mathcal{C}_{\mathsf{L}_{\Psi_q}}(r, x) = - \sum_{y \in \mathcal{Y}} p(y|x) \sum_{j \in [n_e]} \overline{c}_j(x, y) \log\left(\frac{e^{r(x,j)}}{\sum_{j' \in [n_e]} e^{r(x,j')}}\right) = - \sum_{j \in [n_e]} \log(\mathcal{S}(x, j)) \overline{q}(x, j).$$

The conditional regret can be written as

$$\Delta \mathcal{C}_{\mathsf{L}_{\Psi_q}, \mathcal{R}}(r, x)$$

$$= - \sum_{j \in [n_e]} \log(\mathcal{S}(x, j)) \overline{q}(x, j) - \inf_{r \in \mathcal{R}}\left(- \sum_{j \in [n_e]} \log(\mathcal{S}(x, j)) \overline{q}(x, j)\right)$$

$$\geq - \sum_{j \in [n_e]} \log(\mathcal{S}(x, j)) \overline{q}(x, j) - \inf_{\mu \in [-\mathcal{S}(x, j_{\min}(x)), \mathcal{S}(x, \mathsf{r}(x))]}\left(- \sum_{j \in [n_e]} \log(\mathcal{S}_\mu(x, j)) \overline{q}(x, j)\right),$$

where for any $x \in \mathcal{X}$ and $j \in [n_e]$,

$$\mathcal{S}_\mu(x, j) = \begin{cases} \mathcal{S}(x, j), & j \notin \{j_{\min}(x), \mathsf{r}(x)\} \\ \mathcal{S}(x, j_{\min}(x)) + \mu & j = \mathsf{r}(x) \\ \mathcal{S}(x, \mathsf{r}(x)) - \mu & j = j_{\min}(x). \end{cases}$$

Note that with such a choice of $\mathcal{S}_\mu$ leads to the following equality holds:

$$\sum_{j \notin \{\mathsf{r}(x), j_{\min}(x)\}} \log(\mathcal{S}(x, j)) \overline{q}(x, j) = \sum_{j \notin \{\mathsf{r}(x), j_{\min}(x)\}} \log(\mathcal{S}_\mu(x, j)) \overline{q}(x, j).$$

Therefore, the conditional regret of the surrogate loss can be lower bounded as follows:

$$
\Delta \mathcal{C}_{\mathsf{L}_{\Psi_q}, \mathcal{R}}(r, x)
$$

$$
\geq \sup_{\mu \in [-\mathcal{S}(x, j_{\min}(x)), \mathcal{S}(x, r(x))]} \left\{ \overline{q}(x, j_{\min}(x)) [-\log(\mathcal{S}(x, j_{\min}(x))) + \log(\mathcal{S}(x, r(x)) - \mu)] \right.
$$

$$
\left. + \overline{q}(x, r(x)) [-\log(\mathcal{S}(x, r(x))) + \log(\mathcal{S}(x, j_{\min}(x)) + \mu)] \right\}.
$$

By leveraging the concavity of the function, we differentiate it with respect to $\mu$ and set the differential equal to zero to find the maximizing value

$$
\mu^* = \frac{\overline{q}(x, r(x)) \mathcal{S}(x, r(x)) - \overline{q}(x, j_{\min}(x)) \mathcal{S}(x, j_{\min}(x))}{\overline{q}(x, j_{\min}(x)) + \overline{q}(x, r(x))}.
$$

Plugging in the expression of $\mu^*$, we obtain

$$
\Delta \mathcal{C}_{\mathsf{L}_{\Psi_q}, \mathcal{R}}(r, x)
$$

$$
\geq \overline{q}(x, j_{\min}(x)) \log \frac{(\mathcal{S}(x, r(x)) + \mathcal{S}(x, j_{\min}(x))) \overline{q}(x, j_{\min}(x))}{\mathcal{S}(x, j_{\min}(x)) (\overline{q}(x, j_{\min}(x)) + \overline{q}(x, r(x)))}
$$

$$
+ \overline{q}(x, r(x)) \log \frac{(\mathcal{S}(x, r(x)) + \mathcal{S}(x, j_{\min}(x))) \overline{q}(x, r(x))}{\mathcal{S}(x, r(x)) (\overline{q}(x, j_{\min}(x)) + \overline{q}(x, r(x)))}
$$

$$
\geq \overline{q}(x, j_{\min}(x)) \log \frac{2 \overline{q}(x, j_{\min}(x))}{\overline{q}(x, j_{\min}(x)) + \overline{q}(x, r(x))} + \overline{q}(x, r(x)) \log \frac{2 \overline{q}(x, r(x))}{\overline{q}(x, j_{\min}(x)) + \overline{q}(x, r(x))}
$$

$$
\text{(minimum is achieved when } \mathcal{S}(x, r(x)) = \mathcal{S}(x, j_{\min}(x)))
$$

$$
\geq \frac{(\overline{q}(x, r(x)) - \overline{q}(x, j_{\min}(x)))^2}{2(\overline{q}(x, r(x)) + \overline{q}(x, j_{\min}(x)))} \quad \left( a \log \frac{2a}{a+b} + b \log \frac{2b}{a+b} \geq \frac{(a-b)^2}{2(a+b)}, \forall a, b \in [0, 1] \text{ (Mohri et al., 2018, Proposition E.7)} \right)
$$

$$
\geq \frac{(\overline{q}(x, r(x)) - \overline{q}(x, j_{\min}(x)))^2}{4 \overline{C}}. \quad (0 \leq \overline{q}(x, r(x)) + \overline{q}(x, j_{\min}(x)) \leq 2 \overline{C})
$$

Therefore, by Lemma C.1, the conditional regret of the two-stage deferral loss function can be upper bounded as follows:

$$
\Delta \mathcal{C}_{\mathsf{L}_{\text{tdef}}, \mathcal{R}}(r, x) = \overline{q}(x, j_{\min}(x)) - \overline{q}(x, r(x)) \leq 2 (\overline{C})^{\frac{1}{2}} (\Delta \mathcal{C}_{\mathsf{L}_{\Psi_q}, \mathcal{R}}(r, x))^{\frac{1}{2}}.
$$

By the concavity, taking expectations on both sides of the preceding equation, we obtain

$$
\mathcal{E}_{\mathsf{L}_{\text{tdef}}}(r) - \mathcal{E}^*_{\mathsf{L}_{\text{tdef}}}(\mathcal{R}) + \mathcal{M}_{\mathsf{L}_{\text{tdef}}}(\mathcal{R}) \leq 2 (\overline{C})^{\frac{1}{2}} \left( \mathcal{E}_{\mathsf{L}_{\Psi_q}}(r) - \mathcal{E}^*_{\mathsf{L}_{\Psi_q}}(\mathcal{R}) + \mathcal{M}_{\mathsf{L}_{\Psi_q}}(\mathcal{R}) \right)^{\frac{1}{2}}.
$$

**Case II: $q \in (0, 1)$.** In this case, for the surrogate loss $\mathsf{L}_{\Psi_q}$, the conditional error can be written as follows:

$$
q \mathcal{C}_{\mathsf{L}_{\Psi_q}}(r, x) = \sum_{y \in \mathcal{Y}} p(y|x) \sum_{j \in [n_e]} \overline{c}_j(x, y) \left( 1 - \left( \frac{e^{r(x, j)}}{\sum_{j' \in [n_e]} e^{r(x, j')}} \right)^q \right) = \sum_{j \in [n_e]} (1 - \mathcal{S}^q(x, j)) \overline{q}(x, j).
$$

The conditional regret can be written as

$$
q \Delta \mathcal{C}_{\mathsf{L}_{\Psi_q}, \mathcal{R}}(r, x)
$$

$$
= \sum_{j \in [n_e]} (1 - \mathcal{S}^q(x, j)) \overline{q}(x, j) - \inf_{r \in \mathcal{R}} \left( \sum_{j \in [n_e]} (1 - \mathcal{S}^q(x, j)) \overline{q}(x, j) \right)
$$

$$
\geq \sum_{j \in [n_e]} (1 - \mathcal{S}^q(x, j)) \overline{q}(x, j) - \inf_{\mu \in [\mathcal{S}(x, j_{\min}(x)), \mathcal{S}(x, r(x))]} \left( \sum_{j \in [n_e]} (1 - \mathcal{S}^q_\mu(x, j)) \overline{q}(x, j) \right),
$$

where for any $x \in \mathcal{X}$ and $j \in [n_e]$,

$$\mathcal{S}_\mu(x,j) = \begin{cases} \mathcal{S}(x,j), & j \notin \{j_{\min}(x), \mathsf{r}(x)\} \\ \mathcal{S}(x, j_{\min}(x)) + \mu & j = \mathsf{r}(x) \\ \mathcal{S}(x, \mathsf{r}(x)) - \mu & j = j_{\min}(x). \end{cases}$$

Note that with such a choice of $\mathcal{S}_\mu$ leads to the following equality holds:

$$\sum_{j \notin \{\mathsf{r}(x), j_{\min}(x)\}} (1 - \mathcal{S}^q(x,j)) \overline{q}(x,j) = \sum_{j \notin \{\mathsf{r}(x), j_{\min}(x)\}} (1 - \mathcal{S}_\mu^q(x,j)) \overline{q}(x,j).$$

Therefore, the conditional regret of the surrogate loss can be lower bounded as follows:

$$q \Delta \mathcal{C}_{\mathsf{L}_{\Psi_q}, \mathcal{R}}(r, x)$$

$$\geq \sup_{\mu \in [-\mathcal{S}(x, j_{\min}(x)), \mathcal{S}(x, \mathsf{r}(x))]} \left\{ \overline{q}(x, j_{\min}(x)) [-\mathcal{S}^q(x, j_{\min}(x)) + \mathcal{S}^q(x, \mathsf{r}(x)) - \mu] \right.$$

$$\left. + \overline{q}(x, \mathsf{r}(x)) [-\mathcal{S}^q(x, \mathsf{r}(x)) + \mathcal{S}^q(x, j_{\min}(x)) + \mu] \right\}.$$

By leveraging the concavity of the function, we differentiate it with respect to $\mu$ and set the differential equal to zero to find the maximizing value

$$\mu^* = \frac{\overline{q}(x, \mathsf{r}(x))^{\frac{1}{1-q}} \mathcal{S}(x, \mathsf{r}(x)) - \overline{q}(x, j_{\min}(x))^{\frac{1}{1-q}} \mathcal{S}(x, j_{\min}(x))}{\overline{q}(x, j_{\min}(x))^{\frac{1}{1-q}} + \overline{q}(x, \mathsf{r}(x))^{\frac{1}{1-q}}}.$$

Plugging in the expression of $\mu^*$, we obtain

$$q \Delta \mathcal{C}_{\mathsf{L}_{\Psi_q}, \mathcal{R}}(r, x)$$

$$\geq (\mathcal{S}(x, \mathsf{r}(x)) + \mathcal{S}(x, j_{\min}(x)))^q \left( \overline{q}(x, j_{\min}(x))^{\frac{1}{1-q}} + \overline{q}(x, \mathsf{r}(x))^{\frac{1}{1-q}} \right)^{1-q}$$

$$\quad - \overline{q}(x, j_{\min}(x)) \mathcal{S}^q(x, j_{\min}(x)) - \overline{q}(x, \mathsf{r}(x)) \mathcal{S}^q(x, \mathsf{r}(x))$$

$$\geq \frac{1}{n_e{}^q} \left[ 2^q \left( \overline{q}(x, j_{\min}(x))^{\frac{1}{1-q}} + \overline{q}(x, \mathsf{r}(x))^{\frac{1}{1-q}} \right)^{1-q} - \overline{q}(x, j_{\min}(x)) - \overline{q}(x, \mathsf{r}(x)) \right]$$

$$\text{(minimum is achieved when } \mathcal{S}(x, \mathsf{r}(x)) = \mathcal{S}(x, j_{\min}(x)) = \tfrac{1}{n_e})$$

$$\geq \frac{q(\overline{q}(x, \mathsf{r}(x)) - \overline{q}(x, j_{\min}(x)))^2}{4 n_e{}^q \overline{C}}. \qquad \left( \left( \frac{a^{\frac{1}{1-q}} + b^{\frac{1}{1-q}}}{2} \right)^{1-q} - \frac{a+b}{2} \geq \frac{q}{4}(a-b)^2, \forall a, b \in [0,1], 0 \leq a + b \leq 1 \right)$$

Therefore, by Lemma C.1, the conditional regret of the two-stage deferral loss function can be upper bounded as follows:

$$\Delta \mathcal{C}_{\mathsf{L}_{\mathrm{tdef}}, \mathcal{R}}(r, x) = \overline{q}(x, j_{\min}(x)) - \overline{q}(x, \mathsf{r}(x)) \leq 2(n_e{}^q)^{\frac{1}{2}} (\overline{C})^{\frac{1}{2}} \left( \Delta \mathcal{C}_{\mathsf{L}_{\Psi_q}, \mathcal{R}}(r, x) \right)^{\frac{1}{2}}.$$

By the concavity, taking expectations on both sides of the preceding equation, we obtain

$$\mathcal{E}_{\mathsf{L}_{\mathrm{tdef}}}(r) - \mathcal{E}^*_{\mathsf{L}_{\mathrm{tdef}}}(\mathcal{R}) + \mathcal{M}_{\mathsf{L}_{\mathrm{tdef}}}(\mathcal{R}) \leq 2(n_e{}^q)^{\frac{1}{2}} (\overline{C})^{\frac{1}{2}} \left( \mathcal{E}_{\mathsf{L}_{\Psi_q}}(r) - \mathcal{E}^*_{\mathsf{L}_{\Psi_q}}(\mathcal{R}) + \mathcal{M}_{\mathsf{L}_{\Psi_q}}(\mathcal{R}) \right)^{\frac{1}{2}}.$$

$$\qquad \qquad \qquad \qquad \qquad \qquad \qquad \qquad \qquad \qquad \qquad \qquad \qquad \qquad \Box$$

*Proof.* **Case III: $q = 1$.** In this case, for the surrogate loss $\mathsf{L}_{\Psi_q}$, the conditional error can be written as follows:

$$\mathcal{C}_{\mathsf{L}_{\Psi_q}}(r, x) = \sum_{y \in \mathcal{Y}} p(y|x) \sum_{j \in [n_e]} \overline{c}_j(x, y) \left( 1 - \frac{e^{r(x,j)}}{\sum_{j' \in [n_e]} e^{r(x,j')}} \right) = \sum_{j \in [n_e]} (1 - \mathcal{S}(x,j)) \overline{q}(x,j).$$

The conditional regret can be written as

$$\Delta \mathcal{C}_{\mathsf{L}_{\Psi_q}, \mathcal{R}}(r, x)$$

$$= \sum_{j \in [n_e]} (1 - \mathcal{S}(x, j)) \overline{q}(x, j) - \inf_{r \in \mathcal{R}} \left( \sum_{j \in [n_e]} (1 - \mathcal{S}(x, j)) \overline{q}(x, j) \right)$$

$$\geq \sum_{j \in [n_e]} (1 - \mathcal{S}(x, j)) \overline{q}(x, j) - \inf_{\mu \in [\mathcal{S}(x, j_{\min}(x)), \mathcal{S}(x, \mathsf{r}(x))]} \left( \sum_{j \in [n_e]} (1 - \mathcal{S}_{\mu}(x, j)) \overline{q}(x, j) \right),$$

where for any $x \in \mathcal{X}$ and $j \in [n_e]$,

$$\mathcal{S}_{\mu}(x, j) = \begin{cases} \mathcal{S}(x, j), & j \notin \{j_{\min}(x), \mathsf{r}(x)\} \\ \mathcal{S}(x, j_{\min}(x)) + \mu & j = \mathsf{r}(x) \\ \mathcal{S}(x, \mathsf{r}(x)) - \mu & j = j_{\min}(x). \end{cases}$$

Note that with such a choice of $\mathcal{S}_{\mu}$ leads to the following equality holds:

$$\sum_{j \notin \{\mathsf{r}(x), j_{\min}(x)\}} (1 - \mathcal{S}(x, j)) \overline{q}(x, j) = \sum_{j \notin \{\mathsf{r}(x), j_{\min}(x)\}} (1 - \mathcal{S}_{\mu}(x, j)) \overline{q}(x, j).$$

Therefore, the conditional regret of the surrogate loss can be lower bounded as follows:

$$\Delta \mathcal{C}_{\mathsf{L}_{\Psi_q}, \mathcal{R}}(r, x)$$

$$\geq \sup_{\mu \in [-\mathcal{S}(x, j_{\min}(x)), \mathcal{S}(x, \mathsf{r}(x))]} \left\{ \overline{q}(x, j_{\min}(x)) [-\mathcal{S}(x, j_{\min}(x)) + \mathcal{S}(x, \mathsf{r}(x)) - \mu] \right.$$

$$\left. + \overline{q}(x, \mathsf{r}(x)) [-\mathcal{S}(x, \mathsf{r}(x)) + \mathcal{S}(x, j_{\min}(x)) + \mu] \right\}.$$

By leveraging the concavity of the function, we differentiate it with respect to $\mu$ and set the differential equal to zero to find the maximizing value

$$\mu^* = -\mathcal{S}(x, j_{\min}(x)).$$

Plugging in the expression of $\mu^*$, we obtain

$$\Delta \mathcal{C}_{\mathsf{L}_{\Psi_q}, \mathcal{R}}(r, x)$$

$$\geq \overline{q}(x, j_{\min}(x)) \mathcal{S}(x, \mathsf{r}(x)) - \overline{q}(x, \mathsf{r}(x)) \mathcal{S}(x, \mathsf{r}(x))$$

$$\geq \frac{1}{n_e} (\overline{q}(x, \mathsf{r}(x)) - \overline{q}(x, j_{\min}(x))). \qquad \text{(minimum is achieved when } \mathcal{S}(x, \mathsf{r}(x)) = \frac{1}{n_e})$$

Therefore, by Lemma C.1, the conditional regret of the two-stage deferral loss function can be upper bounded as follows:

$$\Delta \mathcal{C}_{\mathsf{L}_{\mathrm{tdef}}, \mathcal{R}}(r, x) = \overline{q}(x, j_{\min}(x)) - \overline{q}(x, \mathsf{r}(x)) \leq n_e (\Delta \mathcal{C}_{\mathsf{L}_{\Psi_q}, \mathcal{R}}(r, x)).$$

By the concavity, taking expectations on both sides of the preceding equation, we obtain

$$\mathcal{E}_{\mathsf{L}_{\mathrm{tdef}}}(r) - \mathcal{E}^*_{\mathsf{L}_{\mathrm{tdef}}}(\mathcal{R}) + \mathcal{M}_{\mathsf{L}_{\mathrm{tdef}}}(\mathcal{R}) \leq n_e \left( \mathcal{E}_{\mathsf{L}_{\Psi_q}}(r) - \mathcal{E}^*_{\mathsf{L}_{\Psi_q}}(\mathcal{R}) + \mathcal{M}_{\mathsf{L}_{\Psi_q}}(\mathcal{R}) \right).$$

$$\square$$

# D. Enhanced bounds under low-noise assumptions: proofs

## D.1. Single-stage: proofs

### D.1.1. PROOF OF LEMMA 5.2

**Lemma 5.2.** *The single-stage deferral Tsybakov noise assumption implies that there exists a constant $c = \frac{B^{1-\alpha}}{\alpha^\alpha} > 0$ such that the following inequalities hold for any $h \in \mathcal{H}$:*

$$\mathbb{E}[1_{\mathsf{h}(x) \neq \mathsf{h}^*(x)}] \leq c \, \mathbb{E}[\gamma(X) 1_{\mathsf{h}(x) \neq \mathsf{h}^*(x)}]^\alpha \leq c [\mathcal{E}_{\mathsf{L}_{\mathrm{def}}}(h) - \mathcal{E}_{\mathsf{L}_{\mathrm{def}}}(h^*)]^\alpha.$$

*Proof.* The second inequality follows directly from the definition of $\gamma(x)$ and Lemma 5.1. The proof of the first inequality follows the same steps as the first part of the proof of Lemma 18 in (Mao et al., 2024e). □

### D.1.2. PROOF OF THEOREM 5.3

We will leverage the following result of Mao et al. (2024e), which serves as a general tool for obtaining enhanced bounds under low-noise assumptions.

**Theorem D.1.** *Assume that there exist two positive functions* $\alpha\colon \mathcal{H} \times \mathcal{X} \to \mathbb{R}_+^*$ *and* $\beta\colon \mathcal{H} \times \mathcal{X} \to \mathbb{R}_+^*$ *with* $\sup_{x \in \mathcal{X}} \alpha(h, x) < +\infty$ *and* $\mathbb{E}_{x \in \mathcal{X}}[\beta(h, x)] < +\infty$ *for all* $h \in \mathcal{H}$ *such that the following holds for all* $h \in \mathcal{H}$ *and* $x \in \mathcal{X}$: $\frac{\Delta\mathcal{C}_{\mathsf{L}_2, \mathcal{H}}(h, x)\, \mathbb{E}_X[\beta(h, x)]}{\beta(h, x)} \leq (\alpha(h, x)\, \Delta\mathcal{C}_{\mathsf{L}_1, \mathcal{H}}(h, x))^{\frac{1}{s}}$, *for some* $s \geq 1$ *with conjugate number* $t \geq 1$, *that is* $\frac{1}{s} + \frac{1}{t} = 1$. *Then, for* $\gamma(h) = \mathbb{E}_X\left[\frac{\alpha^{\frac{t}{s}}(h, x)\beta^t(h, x)}{\mathbb{E}_X[\beta(h, x)]^t}\right]^{\frac{1}{t}}$, *the following inequality holds for any* $h \in \mathcal{H}$:

$$\mathcal{E}_{\mathsf{L}_2}(h) - \mathcal{E}_{\mathsf{L}_2}^*(\mathcal{H}) + \mathcal{M}_{\mathsf{L}_2}(\mathcal{H}) \leq \gamma(h)\big[\mathcal{E}_{\mathsf{L}_1}(h) - \mathcal{E}_{\mathsf{L}_1}^*(\mathcal{H}) + \mathcal{M}_{\mathsf{L}_1}(\mathcal{H})\big]^{\frac{1}{s}}.$$

**Theorem 5.3.** *Consider the setting of single-stage multiple-expert deferral. Assume that the following holds for all* $h \in \mathcal{H}$ *and* $x \in \mathcal{X}$: $\Delta\mathcal{C}_{\mathsf{L}_{\mathrm{def}}, \mathcal{H}}(h, x) \leq \Gamma(\Delta\mathcal{C}_{\mathsf{L}, \mathcal{H}}(h, x))$, *with* $\Gamma(x) = x^{\frac{1}{s}}$, *for some* $s \geq 1$ *with conjugate number* $t \geq 1$, *that is* $\frac{1}{s} + \frac{1}{t} = 1$. *Then, for any* $h \in \mathcal{H}$,

$$\mathcal{E}_{\mathsf{L}_{\mathrm{def}}}(h) - \mathcal{E}_{\mathsf{L}_{\mathrm{def}}}^*(\mathcal{H}) + \mathcal{M}_{\mathsf{L}_{\mathrm{def}}}(\mathcal{H}) \leq \mathbb{E}_X\big[\mathbb{1}_{\mathsf{h}(X) \neq \mathsf{h}^*(X)}\big]^{\frac{1}{t}}\big(\mathcal{E}_{\mathsf{L}}(h) - \mathcal{E}_{\mathsf{L}}^*(\mathcal{H}) + \mathcal{M}_{\mathsf{L}}(\mathcal{H})\big)^{\frac{1}{s}}.$$

*Proof.* Fix $\epsilon > 0$ and define $\beta(h, x) = \mathbb{1}_{\mathsf{h}(x) \neq \mathsf{h}^*(x)} + \epsilon$. By Lemma 5.1,

$$\Delta\mathcal{C}_{\mathsf{L}_{\mathrm{def}}, \mathcal{H}}(h, x) = \max\left\{p_{y_{\max}}, \max_{j \in [n_e]} p_{n+j}\right\} - p_{\mathsf{h}(x)} = p_{\mathsf{h}^*(x)} - p_{\mathsf{h}(x)},$$

we have

$$\frac{\Delta\mathcal{C}_{\mathsf{L}_{\mathrm{def}}, \mathcal{H}}(h, x)\, \mathbb{E}_X[\beta(h, x)]}{\beta(h, x)} \leq \Delta\mathcal{C}_{\mathsf{L}_{\mathrm{def}}, \mathcal{H}}(h, x)\, \mathbb{E}_X[\beta(h, x)],$$

thus the following inequality holds

$$\frac{\Delta\mathcal{C}_{\mathsf{L}_{\mathrm{def}}, \mathcal{H}}(h, x)\, \mathbb{E}_X[\beta(h, x)]}{\beta(h, x)} \leq \mathbb{E}_X[\beta(h, x)]\Delta\mathcal{C}_{\mathsf{L}, \mathcal{H}}^{\frac{1}{s}}(h, x).$$

By Theorem D.1, with $\alpha(h, x) = \mathbb{E}_X[\beta(h, x)]^s$, we have

$$\mathcal{E}_{\mathsf{L}_{\mathrm{def}}}(h) - \mathcal{E}_{\mathsf{L}_{\mathrm{def}}}^*(\mathcal{H}) + \mathcal{M}_{\mathsf{L}_{\mathrm{def}}}(\mathcal{H}) \leq \mathbb{E}_X[\beta^t(h, x)]^{\frac{1}{t}}\big(\mathcal{E}_{\mathsf{L}}(h) - \mathcal{E}_{\mathsf{L}}^*(\mathcal{H}) + \mathcal{M}_{\mathsf{L}}(\mathcal{H})\big)^{\frac{1}{s}}.$$

Since the inequality holds for any $\epsilon > 0$, it implies:

$$\begin{aligned}
\mathcal{E}_{\mathsf{L}_{\mathrm{def}}}(h) - \mathcal{E}_{\mathsf{L}_{\mathrm{def}}}^*(\mathcal{H}) + \mathcal{M}_{\mathsf{L}_{\mathrm{def}}}(\mathcal{H}) &\leq \mathbb{E}_X\Big[\big(\mathbb{1}_{\mathsf{h}(X) \neq \mathsf{h}^*(X)}\big)^t\Big]^{\frac{1}{t}}\big(\mathcal{E}_{\mathsf{L}}(h) - \mathcal{E}_{\mathsf{L}}^*(\mathcal{H}) + \mathcal{M}_{\mathsf{L}}(\mathcal{H})\big)^{\frac{1}{s}} \\
&= \mathbb{E}_X\big[\mathbb{1}_{\mathsf{h}(X) \neq \mathsf{h}^*(X)}\big]^{\frac{1}{t}}\big(\mathcal{E}_{\mathsf{L}}(h) - \mathcal{E}_{\mathsf{L}}^*(\mathcal{H}) + \mathcal{M}_{\mathsf{L}}(\mathcal{H})\big)^{\frac{1}{s}}. \quad \big(\big(\mathbb{1}_{\mathsf{h}(X) \neq \mathsf{h}^*(X)}\big)^t = \mathbb{1}_{\mathsf{h}(X) \neq \mathsf{h}^*(X)}\big)
\end{aligned}$$

This completes the proof. □

### D.1.3. PROOF OF THEOREM 5.4

**Theorem 5.4.** *Consider the setting of single-stage multiple-expert deferral where the Tsybakov noise assumption holds, and no approximation error occurs, that is,* $\mathcal{E}_{\mathsf{L}_{\mathrm{def}}}^*(\mathcal{H}) = \mathcal{E}_{\mathsf{L}_{\mathrm{def}}}^*(\mathcal{H}_{\mathrm{all}})$. *Assume that the following holds for all* $h \in \mathcal{H}$ *and* $x \in \mathcal{X}$: $\Delta\mathcal{C}_{\mathsf{L}_{\mathrm{def}}, \mathcal{H}}(h, x) \leq \Gamma(\Delta\mathcal{C}_{\mathsf{L}, \mathcal{H}}(h, x))$, *with* $\Gamma(x) = x^{\frac{1}{s}}$, *for some* $s \geq 1$. *Then, for any* $h \in \mathcal{H}$,

$$\mathcal{E}_{\mathsf{L}_{\mathrm{def}}}(h) - \mathcal{E}_{\mathsf{L}_{\mathrm{def}}}^*(\mathcal{H}) \leq c^{\frac{s-1}{s - \alpha(s-1)}}\big[\mathcal{E}_{\mathsf{L}}(h) - \mathcal{E}_{\mathsf{L}}^*(\mathcal{H}) + \mathcal{M}_{\mathsf{L}}(\mathcal{H})\big]^{\frac{1}{s - \alpha(s-1)}}.$$

*Proof.* Fix $\epsilon > 0$ and define $\beta(h, x) = 1_{h(x) \neq h^*(x)} + \epsilon$. By Lemma 5.1,

$$\Delta\mathcal{C}_{\mathsf{L}_{\mathrm{def}}, \mathcal{H}}(h, x) = \max\left\{ p_{y_{\max}}, \max_{j \in [n_e]} p_{n+j} \right\} - p_{h(x)} = p_{h^*(x)} - p_{h(x)},$$

we have

$$\frac{\Delta\mathcal{C}_{\mathsf{L}_{\mathrm{def}}, \mathcal{H}}(h, x) \, \mathbb{E}_X[\beta(h, x)]}{\beta(h, x)} \leq \Delta\mathcal{C}_{\mathsf{L}_{\mathrm{def}}, \mathcal{H}}(h, x) \, \mathbb{E}_X[\beta(h, x)],$$

thus the following inequality holds

$$\frac{\Delta\mathcal{C}_{\mathsf{L}_{\mathrm{def}}, \mathcal{H}}(h, x) \, \mathbb{E}_X[\beta(h, x)]}{\beta(h, x)} \leq \mathbb{E}_X[\beta(h, x)] \Delta\mathcal{C}_{\mathsf{L}, \mathcal{H}}^{\frac{1}{s}}(h, x).$$

By Theorem D.1, with $\alpha(h, x) = \mathbb{E}_X[\beta(h, x)]^s$, we have

$$\mathcal{E}_{\mathsf{L}_{\mathrm{def}}}(h) - \mathcal{E}_{\mathsf{L}_{\mathrm{def}}}^*(\mathcal{H}) \leq \mathbb{E}_X[\beta^t(h, x)]^{\frac{1}{t}} \left( \mathcal{E}_{\mathsf{L}}(h) - \mathcal{E}_{\mathsf{L}}^*(\mathcal{H}) + \mathcal{M}_{\mathsf{L}}(\mathcal{H}) \right)^{\frac{1}{s}}.$$

Since the inequality holds for any $\epsilon > 0$, it implies:

$$
\begin{aligned}
\mathcal{E}_{\mathsf{L}_{\mathrm{def}}}(h) - \mathcal{E}_{\mathsf{L}_{\mathrm{def}}}^*(\mathcal{H}) &\leq \mathbb{E}_X\left[ \left( 1_{h(X) \neq h^*(X)} \right)^t \right]^{\frac{1}{t}} \left( \mathcal{E}_{\mathsf{L}}(h) - \mathcal{E}_{\mathsf{L}}^*(\mathcal{H}) + \mathcal{M}_{\mathsf{L}}(\mathcal{H}) \right)^{\frac{1}{s}} \\
&= \mathbb{E}_X[1_{h(X) \neq h^*(X)}]^{\frac{1}{t}} \left( \mathcal{E}_{\mathsf{L}}(h) - \mathcal{E}_{\mathsf{L}}^*(\mathcal{H}) + \mathcal{M}_{\mathsf{L}}(\mathcal{H}) \right)^{\frac{1}{s}} && \left( \left( 1_{h(X) \neq h^*(X)} \right)^t = 1_{h(x) \neq h^*(x)} \right) \\
&\leq c^{\frac{1}{t}} \left[ \mathcal{E}_{\mathsf{L}_{\mathrm{def}}}(h) - \mathcal{E}_{\mathsf{L}_{\mathrm{def}}}^*(\mathcal{H}) \right]^{\frac{\alpha}{t}} \left( \mathcal{E}_{\mathsf{L}}(h) - \mathcal{E}_{\mathsf{L}}^*(\mathcal{H}) + \mathcal{M}_{\mathsf{L}}(\mathcal{H}) \right)^{\frac{1}{s}} && \text{(Tsybakov noise assumption)}
\end{aligned}
$$

The result follows after dividing both sides by $\left[ \mathcal{E}_{\mathsf{L}_{\mathrm{def}}}(h) - \mathcal{E}_{\mathsf{L}_{\mathrm{def}}}^*(\mathcal{H}) \right]^{\frac{\alpha}{t}}$. $\qquad\square$

## D.2. Two-stage: proofs

### D.2.1. LEMMA D.2 AND PROOF

**Lemma D.2.** *The two-stage Tsybakov noise assumption implies that there exists a constant $c = \frac{B^{1-\alpha}}{\alpha^\alpha} > 0$ such that the following inequalities hold for any $h \in \mathcal{R}$:*

$$\mathbb{E}[1_{r(x) \neq r^*(x)}] \leq c \, \mathbb{E}[\gamma(X) 1_{r(x) \neq r^*(x)}]^\alpha \leq c [\mathcal{E}_{\mathsf{L}_{\mathrm{tdef}}}(r) - \mathcal{E}_{\mathsf{L}_{\mathrm{tdef}}}(r^*)]^\alpha.$$

*Proof.* The second inequality follows directly from the definition of $\gamma(x)$ and Lemma 5.1. The proof of the first inequality follows the same steps as the first part of the proof of Lemma 18 in (Mao et al., 2024e). $\qquad\square$

### D.2.2. ENHANCED BOUNDS IN TWO-STAGE SCENARIO (THEOREM D.3 AND THEOREM D.4)

The following result gives an $\mathcal{R}$-consistency bound based on the quantity $1_{r(x) \neq r^*(x)}$.

**Theorem D.3.** *Consider the setting of two-stage multiple-expert deferral. Assume that the following holds for all $h \in \mathcal{R}$ and $x \in \mathcal{X}$: $\Delta\mathcal{C}_{\mathsf{L}_{\mathrm{tdef}}, \mathcal{R}}(r, x) \leq \Gamma(\Delta\mathcal{C}_{\mathsf{L}, \mathcal{R}}(r, x))$, with $\Gamma(x) = x^{\frac{1}{s}}$, for some $s \geq 1$ with conjugate number $t \geq 1$, that is $\frac{1}{s} + \frac{1}{t} = 1$. Then, for any $r \in \mathcal{R}$,*

$$\mathcal{E}_{\mathsf{L}_{\mathrm{tdef}}}(r) - \mathcal{E}_{\mathsf{L}_{\mathrm{tdef}}}^*(\mathcal{R}) + \mathcal{M}_{\mathsf{L}_{\mathrm{tdef}}}(\mathcal{R}) \leq \mathbb{E}_X[1_{r(X) \neq r^*(X)}]^{\frac{1}{t}} \left( \mathcal{E}_{\mathsf{L}}(r) - \mathcal{E}_{\mathsf{L}}^*(\mathcal{R}) + \mathcal{M}_{\mathsf{L}}(\mathcal{R}) \right)^{\frac{1}{s}}.$$

The proof of Theorem D.3 is included in Appendix D.2.3. As noted for a similar result in the single-stage scenario, Theorem D.3 offers a more favorable theoretical guarantee than standard $\mathcal{H}$-consistency bounds, assuming $\Delta\mathcal{C}_{\mathsf{L}_{\mathrm{tdef}}, \mathcal{R}}(r, x) \leq \Gamma(\Delta\mathcal{C}_{\mathsf{L}, \mathcal{R}}(r, x))$.

Next, we assume that the Tsybakov noise assumption holds and also that there is no approximation error, that is $\mathcal{M}_{\mathsf{L}_{\mathrm{def}}}(\mathcal{H}) = 0$.

**Theorem D.4.** *Consider the setting of two-stage multiple-expert deferral where the Tsybakov noise assumption holds, and no approximation error occurs, i.e., $\mathcal{E}^*_{\mathsf{L}_{\mathrm{tdef}}}(\mathcal{R}) = \mathcal{E}^*_{\mathsf{L}_{\mathrm{tdef}}}(\mathcal{R}_{\mathrm{all}})$. Assume that the following holds for all $r \in \mathcal{R}$ and $x \in \mathcal{X}$: $\Delta\mathcal{C}_{\mathsf{L}_{\mathrm{tdef}},\mathcal{R}}(r,x) \leq \Gamma(\Delta\mathcal{C}_{\mathsf{L},\mathcal{R}}(r,x))$, with $\Gamma(x) = x^{\frac{1}{s}}$, for some $s \geq 1$. Then, for any $r \in \mathcal{R}$,*

$$\mathcal{E}_{\mathsf{L}_{\mathrm{tdef}}}(r) - \mathcal{E}^*_{\mathsf{L}_{\mathrm{tdef}}}(\mathcal{R}) \leq c^{\frac{s-1}{s-\alpha(s-1)}}\left[\mathcal{E}_{\mathsf{L}}(r) - \mathcal{E}^*_{\mathsf{L}}(\mathcal{R}) + \mathcal{M}_{\mathsf{L}}(\mathcal{R})\right]^{\frac{1}{s-\alpha(s-1)}}.$$

The proof can be found in Appendix D.1.3. As in the single-stage scenario, in the case of $\alpha \to 1$, where Tsybakov noise corresponds to Massart's noise assumption, this provides an $\mathcal{H}$-consistency bound with a linear functional form, improving upon the standard $\mathcal{H}$-consistency bounds, which have the functional form $\Gamma(x) = x^{\frac{1}{s}}$, if they exist. This demonstrates that for any smooth surrogate loss with an $\mathcal{H}$-consistency bound, under Massart's noise assumption, we can obtain $\mathcal{H}$-consistency bounds as favorable as those of $\mathsf{L} = \mathsf{L}_{\Psi_q}$ with $q = 1$, which always admits a linear dependency, as shown in Theorem 4.4.

For example, Theorem 4.4 shows that square-root $\mathcal{H}$-consistency bounds hold ($s = \frac{1}{2}$) for $\mathsf{L} = \mathsf{L}_{\Psi_q}$ with $q \in [0,1)$. Thus, Theorem D.4 provides refined $\mathcal{H}$-consistency bounds for these surrogate losses: a linear dependence when Massart's noise assumption holds ($\alpha \to 1$) and an intermediate rate between linear and square-root for other values of $\alpha$ within the range $(0,1)$.

Furthermore, the realizability assumption can be viewed as a special case of Massart's noise assumption. Thus, Theorem D.4 provides intermediate guarantees between realizable $\mathcal{H}$-consistency and $\mathcal{H}$-consistency bounds for two-stage deferral surrogate losses, such as $\mathsf{L}_\Phi$ in (5) for the two-expert case and $\mathsf{L}_\Psi$ in (7) for the multiple-expert case.

### D.2.3. PROOF OF THEOREM D.3

**Theorem D.3.** *Consider the setting of two-stage multiple-expert deferral. Assume that the following holds for all $h \in \mathcal{R}$ and $x \in \mathcal{X}$: $\Delta\mathcal{C}_{\mathsf{L}_{\mathrm{tdef}},\mathcal{R}}(r,x) \leq \Gamma(\Delta\mathcal{C}_{\mathsf{L},\mathcal{R}}(r,x))$, with $\Gamma(x) = x^{\frac{1}{s}}$, for some $s \geq 1$ with conjugate number $t \geq 1$, that is $\frac{1}{s} + \frac{1}{t} = 1$. Then, for any $r \in \mathcal{R}$,*

$$\mathcal{E}_{\mathsf{L}_{\mathrm{tdef}}}(r) - \mathcal{E}^*_{\mathsf{L}_{\mathrm{tdef}}}(\mathcal{R}) + \mathcal{M}_{\mathsf{L}_{\mathrm{tdef}}}(\mathcal{R}) \leq \mathbb{E}_X\left[1_{r(X)\neq r^*(X)}\right]^{\frac{1}{t}}\left(\mathcal{E}_{\mathsf{L}}(r) - \mathcal{E}^*_{\mathsf{L}}(\mathcal{R}) + \mathcal{M}_{\mathsf{L}}(\mathcal{R})\right)^{\frac{1}{s}}.$$

*Proof.* Fix $\epsilon > 0$ and define $\beta(r,x) = 1_{r(x)\neq r^*(x)} + \epsilon$. By Lemma C.1,

$$\Delta\mathcal{C}_{\mathsf{L}_{\mathrm{tdef}},\mathcal{R}}(r,x) = \sum_{y\in\mathcal{Y}} p(y|x)c_{r(x)}(x,y) - \sum_{y\in\mathcal{Y}} p(y|x)c_{r^*(x)}(x,y),$$

we have

$$\frac{\Delta\mathcal{C}_{\mathsf{L}_{\mathrm{tdef}},\mathcal{R}}(r,x)\,\mathbb{E}_X[\beta(r,x)]}{\beta(r,x)} \leq \Delta\mathcal{C}_{\mathsf{L}_{\mathrm{tdef}},\mathcal{R}}(r,x)\,\mathbb{E}_X[\beta(r,x)],$$

thus the following inequality holds

$$\frac{\Delta\mathcal{C}_{\mathsf{L}_{\mathrm{tdef}},\mathcal{R}}(r,x)\,\mathbb{E}_X[\beta(r,x)]}{\beta(r,x)} \leq \mathbb{E}_X[\beta(r,x)]\Delta\mathcal{C}^{\frac{1}{s}}_{\mathsf{L},\mathcal{R}}(r,x).$$

By Theorem D.1, with $\alpha(r,x) = \mathbb{E}_X[\beta(r,x)]^s$, we have

$$\mathcal{E}_{\mathsf{L}_{\mathrm{tdef}}}(r) - \mathcal{E}^*_{\mathsf{L}_{\mathrm{tdef}}}(\mathcal{R}) + \mathcal{M}_{\mathsf{L}_{\mathrm{tdef}}}(\mathcal{R}) \leq \mathbb{E}_X[\beta^t(r,x)]^{\frac{1}{t}}\left(\mathcal{E}_{\mathsf{L}}(r) - \mathcal{E}^*_{\mathsf{L}}(\mathcal{R}) + \mathcal{M}_{\mathsf{L}}(\mathcal{R})\right)^{\frac{1}{s}}.$$

Since the inequality holds for any $\epsilon > 0$, it implies:

$$\mathcal{E}_{\mathsf{L}_{\mathrm{tdef}}}(r) - \mathcal{E}^*_{\mathsf{L}_{\mathrm{tdef}}}(\mathcal{R}) + \mathcal{M}_{\mathsf{L}_{\mathrm{tdef}}}(\mathcal{R}) \leq \mathbb{E}_X\left[\left(1_{r(X)\neq r^*(X)}\right)^t\right]^{\frac{1}{t}}\left(\mathcal{E}_{\mathsf{L}}(r) - \mathcal{E}^*_{\mathsf{L}}(\mathcal{R}) + \mathcal{M}_{\mathsf{L}}(\mathcal{R})\right)^{\frac{1}{s}}$$

$$= \mathbb{E}_X\left[1_{r(X)\neq r^*(X)}\right]^{\frac{1}{t}}\left(\mathcal{E}_{\mathsf{L}}(r) - \mathcal{E}^*_{\mathsf{L}}(\mathcal{R}) + \mathcal{M}_{\mathsf{L}}(\mathcal{R})\right)^{\frac{1}{s}}. \quad \left(\left(1_{r(X)\neq r^*(X)}\right)^t = 1_{r(X)\neq r^*(X)}\right)$$

This completes the proof. $\qquad\square$

### D.2.4. PROOF OF THEOREM D.4

**Theorem D.4.** *Consider the setting of two-stage multiple-expert deferral where the Tsybakov noise assumption holds, and no approximation error occurs, i.e., $\mathcal{E}^*_{\mathsf{L}_{\mathrm{tdef}}}(\mathcal{R}) = \mathcal{E}^*_{\mathsf{L}_{\mathrm{tdef}}}(\mathcal{R}_{\mathrm{all}})$. Assume that the following holds for all $r \in \mathcal{R}$ and $x \in \mathcal{X}$:*
$\Delta\mathcal{C}_{\mathsf{L}_{\mathrm{tdef}},\mathcal{R}}(r,x) \leq \Gamma(\Delta\mathcal{C}_{\mathsf{L},\mathcal{R}}(r,x))$, with $\Gamma(x) = x^{\frac{1}{s}}$, for some $s \geq 1$. *Then, for any $r \in \mathcal{R}$,*

$$\mathcal{E}_{\mathsf{L}_{\mathrm{tdef}}}(r) - \mathcal{E}^*_{\mathsf{L}_{\mathrm{tdef}}}(\mathcal{R}) \leq c^{\frac{s-1}{s-\alpha(s-1)}} \big[\mathcal{E}_{\mathsf{L}}(r) - \mathcal{E}^*_{\mathsf{L}}(\mathcal{R}) + \mathcal{M}_{\mathsf{L}}(\mathcal{R})\big]^{\frac{1}{s-\alpha(s-1)}}.$$

*Proof.* Fix $\epsilon > 0$ and define $\beta(r,x) = 1_{\mathsf{r}(x) \neq \mathsf{r}^*(x)} + \epsilon$. By Lemma 5.1,

$$\Delta\mathcal{C}_{\mathsf{L}_{\mathrm{tdef}},\mathcal{R}}(r,x) = \sum_{y \in \mathcal{Y}} p(y|x)c_{\mathsf{r}(x)}(x,y) - \sum_{y \in \mathcal{Y}} p(y|x)c_{\mathsf{r}^*(x)}(x,y),$$

we have

$$\frac{\Delta\mathcal{C}_{\mathsf{L}_{\mathrm{tdef}},\mathcal{R}}(r,x)\,\mathbb{E}_X[\beta(r,x)]}{\beta(r,x)} \leq \Delta\mathcal{C}_{\mathsf{L}_{\mathrm{tdef}},\mathcal{R}}(r,x)\,\mathbb{E}_X[\beta(r,x)],$$

thus the following inequality holds

$$\frac{\Delta\mathcal{C}_{\mathsf{L}_{\mathrm{tdef}},\mathcal{R}}(r,x)\,\mathbb{E}_X[\beta(r,x)]}{\beta(r,x)} \leq \mathbb{E}_X[\beta(r,x)]\Delta\mathcal{C}^{\frac{1}{s}}_{\mathsf{L},\mathcal{R}}(r,x).$$

By Theorem D.1, with $\alpha(r,x) = \mathbb{E}_X[\beta(r,x)]^s$, we have

$$\mathcal{E}_{\mathsf{L}_{\mathrm{tdef}}}(r) - \mathcal{E}^*_{\mathsf{L}_{\mathrm{tdef}}}(\mathcal{R}) \leq \mathbb{E}_X[\beta^t(r,x)]^{\frac{1}{t}}\big(\mathcal{E}_{\mathsf{L}}(r) - \mathcal{E}^*_{\mathsf{L}}(\mathcal{R}) + \mathcal{M}_{\mathsf{L}}(\mathcal{R})\big)^{\frac{1}{s}}.$$

Since the inequality holds for any $\epsilon > 0$, it implies:

$$
\begin{aligned}
\mathcal{E}_{\mathsf{L}_{\mathrm{tdef}}}(r) - \mathcal{E}^*_{\mathsf{L}_{\mathrm{tdef}}}(\mathcal{R}) &\leq \mathbb{E}_X\Big[\big(1_{\mathsf{r}(X) \neq \mathsf{r}^*(X)}\big)^t\Big]^{\frac{1}{t}}\big(\mathcal{E}_{\mathsf{L}}(r) - \mathcal{E}^*_{\mathsf{L}}(\mathcal{R}) + \mathcal{M}_{\mathsf{L}}(\mathcal{R})\big)^{\frac{1}{s}} \\
&= \mathbb{E}_X[1_{\mathsf{r}(X) \neq \mathsf{r}^*(X)}]^{\frac{1}{t}}\big(\mathcal{E}_{\mathsf{L}}(r) - \mathcal{E}^*_{\mathsf{L}}(\mathcal{R}) + \mathcal{M}_{\mathsf{L}}(\mathcal{R})\big)^{\frac{1}{s}} && \Big(\big(1_{\mu(r,X)>0}\big)^t = 1_{\mu(r,X)>0}\Big) \\
&\leq c^{\frac{1}{t}}\big[\mathcal{E}_{\mathsf{L}_{\mathrm{tdef}}}(r) - \mathcal{E}^*_{\mathsf{L}_{\mathrm{tdef}}}(\mathcal{R})\big]^{\frac{\alpha}{t}}\big(\mathcal{E}_{\mathsf{L}}(r) - \mathcal{E}^*_{\mathsf{L}}(\mathcal{R}) + \mathcal{M}_{\mathsf{L}}(\mathcal{R})\big)^{\frac{1}{s}} && \text{(Tsybakov noise assumption)}
\end{aligned}
$$

The result follows after dividing both sides by $\big[\mathcal{E}_{\mathsf{L}_{\mathrm{tdef}}}(r) - \mathcal{E}^*_{\mathsf{L}_{\mathrm{tdef}}}(\mathcal{R})\big]^{\frac{\alpha}{t}}$. $\qquad\square$

