# OpenReview forum: "Mastering Multiple-Expert Routing: Realizable $H$-Consistency and Strong Guarantees for Learning to Defer"
_ICML.cc/2025/Conference — ICML 2025 poster_

### Official Review · Reviewer_m1gt · 2025-03-02

**Overall Recommendation:** 3

**Summary:**

The paper presents a novel surrogate loss for both single-stage and two-stage learning to defer. The authors show several theoretical guarantees, including $H$-consistency, Bayes consistency and $H$-consistency bounds for both the single and multiple expert settings.
Empirical results showcase the effectiveness of the approach.


### Update after the Rebuttal

I am satisfied with the authors' answers. I keep my positive score.

**Claims And Evidence:**

The theoretical contribution is strong, with several new results for Learning to Defer losses.

**Essential References Not Discussed:**

To the best of my knowledge, the paper discusses the most relevant works in the area of learning to defer.

**Experimental Designs Or Analyses:**

See the evaluation criteria shortcomings.

**Methods And Evaluation Criteria:**

The empirical evaluation is limited, considering only a couple of baselines and the multi-expert case.
The choice of datasets could be improved. For example, the authors could look at the empirical evaluations of (Mozannar et al., 2023) or (Palomba et al., 2024), where they consider datasets with real human labels, rather than simulated ones.
Moreover, the authors state that:
> It also defers significantly more and exhibits a greater tendency to choose expert 2 or expert 3 compared to the baselines, demonstrating the effect of our proposed surrogate loss.

 I am not entirely convinced that deferring more to experts is desirable in real-life applications, where querying experts is typically more costly than using an ML model.
For instance, in both references mentioned before, the authors provide accuracy - coverage curves, to show how performance is affected when limiting the ability to defer.

**Other Comments Or Suggestions:**

Please double-check the arXiv papers in the references to see if they have an accepted version of the paper.

**Other Strengths And Weaknesses:**

The paper advances the Learning to Defer literature with a novel surrogate loss family. Therefore, it is incremental in novelty but rigorous and sound in terms of theoretical aspects.

I argue that the paper's main weakness is its limited empirical evaluation (see detailed comments in the other sections here).

However, the merits of the paper still outweigh the current shortcomings, so I am prone to suggesting a weak acceptance.

**Questions For Authors:**

Can the authors briefly comment on my comments in the Theoretical Claims sections and the Methods And Evaluation Criteria?

**Relation To Broader Scientific Literature:**

The related work is adequately discussed in the Appendix section.

**Theoretical Claims:**

I checked most of the proofs, and they seem correct.
However, I have a few observations regarding the notation and how the proofs are presented:
1) some notation choices that are uncommon and a bit cumbersome: e.g., for Theorem 3.1 and Theorem 4.1, denoting the conditional distribution $P(Y=y | X=x)$ as $p(x,y)$  is in my view just misleading as $p(x,y)$ recalls more naturally the joint distribution. I would propose the authors to change notation into something like $p(y|x)$, which is more natural in my opinion;
2) In my opinion, several passages of the proofs are just not adequately commented on. Adding more comments is a courtesy to the reader, which can help make the whole proof more accessible. For example, the equation lines 4 to 6 in the proof of  Lemma 3.1 could be better presented, by adding a bit of comment on the fact you add and subtract $ \sum_{j=1}^{n_e} \mathbb{I}[(h(x) \neq y)] \mathbb{I}[h(x)\neq n+j] $ and then explain why $(1-n_e) = (\prod_{j=1}^{n_e}\mathbb{I}[h(x)\neq n+j] - \sum_{j=1}^{n_e} \mathbb{I}[h(x)\neq n+j]) $.

---

> ### Author Rebuttal · Authors · 2025-03-31
>
> Thank you for your encouraging review. We will take your suggestions into account when preparing the final version. Below please find responses to specific questions.
>
> **1. Empirical Evaluation:** Thank you for the suggestions. As recommended by the reviewer, we will aim to include experiments involving real-world experts, such as those considered in (Mozannar et al., 2023) and (Palomba et al., 2024), in the final version.
>
> But, as you have mentioned, our paper primarily focuses on theoretically principled surrogate losses for learning to defer with multiple experts, grounded in realizable $H$-consistency and $H$-consistency bounds, within the learning theory framework. While we have empirically shown that minimizing our proposed loss functions outperforms the baselines in realizable settings and matches the best-known results in synthetic non-realizable settings—as predicted by our theoretical analysis—we recognize the importance of further empirical exploration. Accordingly, we plan to dedicate future work to conducting a more extensive empirical analysis in real-world settings, particularly in important scenarios where expert domains overlap (as suggested by Reviewer wQs7) and the data is heterogeneous (as suggested by Reviewer 798e).
>
> We will also include a more detailed discussion of the limitations of our empirical analysis and outline potential directions for future work in the final version. Nevertheless, we believe our theoretical contributions are significant and well-suited for acceptance in the theory track of the ICML conference.
>
> **2. Deferring More to Experts:** We agree with the reviewer that deferring more to experts may not always be preferred in real-world applications, where querying experts can be more costly than relying on an ML model. We would like to clarify that the goal of our experiments on non-realizable data is not to claim that our proposed surrogate losses outperform the baselines in such settings. Rather, our results demonstrate that our surrogate losses match the performance of these state-of-the-art baselines in non-realizable scenarios in terms of system accuracy, as both are supported by consistency bounds.
>
> The strength of our approach lies in the realizable setting, where our surrogate losses are guaranteed to be realizable $H$-consistent—unlike the baselines—as illustrated in Figure 1. Due to the specific realizable form of our surrogate losses, our model also defers significantly more and exhibits a greater tendency to select expert 2 compared to the baselines of Verma et al. (2023) and Mao et al. (2024b). This highlights a key effect of our proposed surrogate: it induces a deferral behavior that is qualitatively different from prior approaches.
>
> **3. Theoretical Claims:** Thank you for the suggestions. We will revise the notation for the conditional distribution and add further commentary on the proof based on your feedback.
>
> **4. Other Comments Or Suggestions:** Thank you for the suggestion. We will double-check the arXiv references and update them with their accepted versions, if available, in the final version.

---

### Official Review · Reviewer_A5VC · 2025-03-14

**Overall Recommendation:** 4

**Summary:**

This paper studies 1. single-stage multiple-expert deferral, which is the deferral problem where the learner has the option to predict or defer the prediction to one of pre-defined experts, and 2. two-stage multiple-expert deferral, which assumes a label predictor have been learned in first stage and the learner select one from a set of pre-defined experts. The authors proposed realizable H-consistent surrogate loss for single stage and realizable H-consistent surrogate loss, H-consistency bounds surrogate loss for two stages. The properties of surrogate loss are proposed in a line of previous work.

**Claims And Evidence:**

The paper provides proof for their claims.

**Essential References Not Discussed:**

N/A

**Experimental Designs Or Analyses:**

N/A

**Methods And Evaluation Criteria:**

N/A

**Other Comments Or Suggestions:**

N/A

**Other Strengths And Weaknesses:**

The paper is well written and discusses from both theoretical and practical perspectives. The authors show good properties regarding the loss families they proposed in two problem settings and provide complete analysis under low-noise assumptions.

**Questions For Authors:**

Are there any other interesting properties or improvements the authors want to achieve for their loss family besides those already mentioned in the paper? It would be nice if the authors could discuss a few sentences about future directions.

**Relation To Broader Scientific Literature:**

This seems to be a very interesting learning setting and a good problem to study the properties of loss families.

**Theoretical Claims:**

N/A

---

> ### Author Rebuttal · Authors · 2025-03-31
>
> Thank you for your appreciation of our work. We will take your suggestions into account when preparing the final version. Below please find responses to specific questions.
>
> **1. Questions: Are there any other interesting properties or improvements the authors want to achieve for their loss family besides those already mentioned in the paper? It would be nice if the authors could discuss a few sentences about future directions.**
>
> **Response:** As suggested by Reviewer 798e, extending our methods to more realistic settings—such as those where experts may not be available during training—is a promising direction for future work. For example, it would be interesting to adapt our framework to handle previously unseen experts at test time, as studied by Tailor et al. (2024, AISTATS), and to incorporate post-processing frameworks for learning to defer under constraints such as OOD detection and long-tail classification, as explored by Charusaie et al. (2024, NeurIPS). We will further elaborate on this potential direction in the paper.
>
> We also plan to conduct a more extensive empirical analysis in real-world settings, particularly in important scenarios where expert domains overlap (as suggested by Reviewer wQs7) and where the data is heterogeneous (as suggested by Reviewer 798e).
>
> [1] Tailor et al. (2024). Learning to Defer to a Population: A Meta-Learning Approach. AISTATS 2024.
>
> [2] Charusaie et al. (2024). A Unifying Post-Processing Framework for Multi-Objective Learn-to-Defer Problems. NeurIPS 2024.

---

### Official Review · Reviewer_798e · 2025-03-14

**Overall Recommendation:** 3

**Summary:**

This work introduces a family of surrogate losses functions for the learning to defer. The authors discussed methods for achieving H-consistent loss for single-stage deferral, and H-consistency and Bayes consistency for two-/multi-expert scenarios. The authors propose to surrogate the undifferentiable indicator functions in classic learning to defer losses with a family of comp-sum losses (a general case of the loss proposed by Sontag et al. [1]). The authors then discuss the the realizable H-consistency and H-consistency bounds for special cases of comp-sum losses for single-stage multi-expert deferral and two-stage multiple-expert deferrals. The authors then discuss the assumptions with which H-consistency bounds can be achieved. The empirical performance of the proposed family of surrogate loss are evaluated four public datasets.The proposed method yields improved system accuracy compared to existing works.

[1] Consistent Estimators for Learning to Defer to an Expert

## update after rebuttal

I would like to thank the authors for the clarifications. While I am still not fully convinced on the realism of problem setting, I would like to increase the score in light of the contributions over existing learning to defer works.

**Claims And Evidence:**

The authors claim improved results compared with some existing baselines, which can be partly verified in the experiments in terms of SA.
The authors claim to have achieved H-consistency for single-stage deferral and realizable H-consistency, H-consistency bounds, and Bayes-consistency for the two-/multiple-expert scenario. This is presented with the derivations in the manuscript and supplementary results.

**Essential References Not Discussed:**

In practice, methods for allowing a model deferring decisions to external experts are many, e.g., those based on uncertainty estimation or anomaly detection. The authors are encouraged to briefly talk about these related problems.

**Experimental Designs Or Analyses:**

Although improved SA's are achieved, I have concerns on the realism of experiments: The employed datasets are mostly toy datasets which may not represent the complexity of real world heterogeneous data (e.g., more complicated tasks, coupling with class imbalance and OOD scenarios). Therefore, the real-world value of the proposed work is unclear to me when compared with previous learning to defer methods.
In terms of realism another issue is that getting sufficient samples on characterizing the correctness of experts at the system building stage a may be difficult -- it is often hindered by data / expert availability.

**Methods And Evaluation Criteria:**

The methodology is straightforward (essentially an extension to [1]) and the use of SA for assessing the deferral performance seems to be reasonable.

**Other Comments Or Suggestions:**

N/A

**Other Strengths And Weaknesses:**

Strengths:
- The manuscript is built on substantial theoretical derivation.
- The methodology is simple.
- Improved SA has been achieved on four datasets.

Weaknesses:
- Realism in problem setting: In practice it is often very unlikely for a system to be able to incorporate deferral mechanism to a set of fixed experts (not to mention that sample predictions of experts or experts themselves are often unavailable at the time a predictor is built). Often times, human experts are even not always stationary. In this sense methods based on uncertainty estimation / anomaly detection are way more practical (especially for post-hoc ones).

- Realism in datasets: The experimented datasets seem to be over-simplistic compared to real-world heterogeneous data, which are way more difficult to learn and are often inevitably noisy, class-imbalanced, and may contain OOD samples (which can easily overshadow the advantage brought by having H-consistent training losses).

- Despite the authors hint a few times the significance of learning to defer for large language models, throughout the paper, there is little discussions on how the proposed loss would be integrated to LLMs nor are there experiments supporting it.

**Questions For Authors:**

While I appreciate the efforts on pushing forward in approaching H-consistency of losses, my major question is on the realism of the problem setting and over-simplified experiment substantiations. These issues may incur doubts on the real-world value of the proposed work.

**Relation To Broader Scientific Literature:**

This work belongs to the family of learning to defer problems. However, in practice, methods for allowing a model deferring decisions to external experts are many, e.g., those based on uncertainty estimation or anomaly detection. Many of them have the advantage that they do not to need to take experts (and their prediction examples) into consideration when a predictor is built, which bestow them significantly more practical value compared to the learning to defer methods.

**Theoretical Claims:**

I have not identified significant issues with theoretical claims.

---

> ### Author Rebuttal · Authors · 2025-03-31
>
> Thank you for your insightful comments. We will take your suggestions into account when preparing the final version. Below please find responses to specific questions.
>
> **1. Experimental Designs Or Analyses:** Thank you for the suggestions. As recommended by Reviewer m1gt, we will aim to include experiments involving real-world experts, such as those considered in (Mozannar et al., 2023) and (Palomba et al., 2024), in the final version.
>
> Our paper primarily focuses on theoretically principled surrogate losses for learning to defer with multiple experts, grounded in realizable $H$-consistency and $H$-consistency bounds, within the learning theory framework. While we have empirically shown that minimizing our proposed loss functions outperforms the baselines in realizable settings and matches the best-known results in synthetic non-realizable settings—as predicted by our theoretical analysis—we recognize the importance of further empirical exploration. Accordingly, we plan to dedicate future work to conducting a more extensive empirical analysis in real-world settings, particularly in important scenarios where expert domains overlap (as suggested by Reviewer wQs7) and the data is heterogeneous (as suggested by Reviewer 798e).
>
> We will also include a more detailed discussion of the limitations of our empirical analysis and outline potential directions for future work in the final version. Nevertheless, we believe our theoretical contributions are significant and well-suited for acceptance in the theory track of the ICML conference.
>
> **2. Relation To Broader Scientific Literature:** As the reviewer suggests, we are happy to include an additional discussion of methods based on uncertainty estimation. Previous work has shown that some of these methods can be suboptimal compared to those based on deferral or abstention, as studied in our paper. The reviewer references “uncertainty estimation” and “anomaly detection” methods multiple times; if there are specific publications they recommend, we would gladly discuss them in the final version.
>
> **3. Problem Setting:** We would like to point out that the same problem setting has been considered in several notable previous works on learning to defer (Mozannar & Sontag, 2020; Verma & Nalisnick, 2022; Charusaie et al., 2022; Mozannar et al., 2023; Verma et al., 2023; Mao et al., 2024b, among others). Within this setting, our paper introduces novel surrogate loss functions and efficient algorithms with strong theoretical learning guarantees. We address open questions regarding realizable $H$-consistency, $H$-consistency bounds, and Bayes consistency for both the single-stage setting (jointly learning the predictor and deferral function) and the two-stage setting (learning only the deferral function with a fixed expert).
>
> We acknowledge that more realistic settings—such as those where experts may not be available during training, as mentioned by the reviewer—are also important. Extending our methods to these scenarios is a promising direction for future work. For example, it would be interesting to adapt our framework to handle previously unseen experts at test time, as studied by Tailor et al. (2024, AISTATS), and to incorporate post-processing frameworks for learning to defer under constraints such as OOD detection and long-tail classification, as explored by Charusaie et al. (2024, NeurIPS). We will further elaborate on this potential direction in the paper.
>
> [1] Tailor et al. (2024). Learning to Defer to a Population: A Meta-Learning Approach. AISTATS 2024.
>
> [2] Charusaie et al. (2024). A Unifying Post-Processing Framework for Multi-Objective Learn-to-Defer Problems. NeurIPS 2024.
>
> **4. Application to Large Language Models:** Our deferral solution can indeed provide a useful solution in the context of LLMs. Given the significant resources required to retrain LLMs, a two-stage method is a practical and scalable solution. This does not require modifying LLMs or the loss function used to train them.
>
> Our proposed two-stage surrogate loss framework is particularly well-suited for use with pre-trained LLMs. In the presence of multiple LLMs (or experts), using our method, uncertain predictions can be deferred to more specialized or domain-specific pre-trained models, thereby both improving the reliability and accuracy of the overall system and improving efficiency. We will further elaborate on this in the final version.
>
> We have in fact carried out preliminary experiments demonstrating the effectiveness of our deferral method in enhancing LLM performance. However, these results are currently not reported publicly due to IP restrictions.

---

### Official Review · Reviewer_wQs7 · 2025-03-16

**Overall Recommendation:** 3

**Summary:**

In the paper  _Mastering Multiple-Expert Routing: Realizable-Consistency and Strong Guarantees for Learning to Defer_, authors propose surrogate losses for learning to defer that satisfy the following oints: realizable H-consistency, H-consistency bounds, and Bayes consistency
for both single-stage and two-stage learning scenarios. The authors mathematically prove that their claims and compare their proposed solution to some baselines for a few common datasets.

**Claims And Evidence:**

In Lines 70-74, first column, the authors claim that  in  ‘*multiple-expert case, Verma et al.(2023) extended the surrogate loss proposed in (Verma & Nalisnick, 2022), and Mao et al. (2024b) extended the surrogate loss introduced in (Mozannar & Sontag, 2020) to the multiple-expert setting’.* Verma et al.(2023) also extended (Mozannar & Sontag, 2020) to the multiple expert setup, proving Bayes consistency. This claim from the authors throughout this paragraph is quite strong, and the authors are failing to acknowledge the contributions from Verma et al.(2023), the first work to extend *(Verma* & Nalisnick, 2022) and (*Mozannar & Sontag, 2020*) surrogate losses in L2D to multiple experts (single stage).

This claim is also repeated in Appendix A, lines 750 - 755.

**Essential References Not Discussed:**

All main references for learning to defer are mentioned.

**Experimental Designs Or Analyses:**

The experimental analysis seems very concise. While the authors compare to a few baselines using three different datasets, I think the authors limit the experimentation to a very ideal scenario where experts’ domain are independent.

While this assumption could make sense when using pretrained ML models as predictors (for example), for a real use case with actual human experts, it’s rarely the case that experts predict completely independent. That is, even if a doctor is an expert in a set of classes in CIFAR-10 (e.g. bird, cat), that expert most likely would also predict with a high probability for other classes (e.g. dog, deer, frog, etc., because these are also animals). This results in a more difficult scenario which has not been explored in this work.

**Methods And Evaluation Criteria:**

#

1. The new proposed surrogate losses for the multiple-expert setup are tested only with a few number of experts (up to three experts).
2. So far in the experiments, the authors only addressed the case where the knowledge of the experts is complementary. That is, each expert has a specific expertise w.r.t to some classes,e.g. in the single-stage multiple expert deferral one expert is oracle on the first 30% classes, and wrong otherwise. Second expert is the complement of the first, predicting always correct for the next 30% of the classes, and wrong otherwise. As studied in Verma et al., 2023, the application of deferral-based mechanisms as studied here becomes more challenging when the experts’ domains overlap. I think this case is not yet addresses in this paper. How would the proposed solution work under experts’ domain overlap?

---

Verma, R., Barrejón, D. and Nalisnick, E., 2023, April. Learning to defer to multiple experts: Consistent surrogate losses, confidence calibration, and conformal ensembles. In *International Conference on Artificial Intelligence and Statistics* (pp. 11415-11434). PMLR.

**Other Comments Or Suggestions:**

I think introduction section is quite long - specially the enumeration of the different sections and its content. This could be shortened. But just a suggestion. Maybe leave more space for real-life use case and motivating examples to allow non-experts on the field to understand the learning to defer paradigm. ƒ

**Other Strengths And Weaknesses:**

It’s clear that the authors of the paper are aware of the recent papers in the learning to defer literature, and are worried about proposing actual consistent surrogate losses. Because of this, the proposed solutions are theoretically proved and guaranteed.

Regarding the weaknesses, I think the expert domain probabilities definition for the synthetical experts is quite ideal, where the expert domain is independent among experts. This is not commented in the paper, and hence not addressed, when this is an actual real scenario that would happen in real life.

**Questions For Authors:**

1. In Figure 1, the authors show the outperformance of the proposed solution choosing $q=0.7$ and $q=1$. How would it behave for lower values? Could you include such results? Even if results are worse, it would show the limitations of the work.
    1. How does the contribution actually defer from previous works (e.g. Verma et al., 2023, Mao et al., 2024b) in terms of performance, i.e. system accuracy?
2. From Table 1, it seems that for single-stage baseline, the proposed solution is not so far from the compared baselines. It’s also quite surprising that for Verma et al., 2023, the deferral ratio for the two experts for SVHN, for the second expert is rather low. Could you comment why it’s that the case?

**Relation To Broader Scientific Literature:**

The work is mainly related to learning to defer literature and deferral-based methods. The paper does not really mention or emphasize the importance of such deferral-based mechanisms in real world machine-collaboration scenarios. It excels in its mathematical contributions.

**Theoretical Claims:**

All claims seem to be mathematically grounded, although I didn't dive too deep into the proofs and derivations in the Appendix.

---

> ### Author Rebuttal · Authors · 2025-03-31
>
> Thank you for your encouraging review. We will take your suggestions into account when preparing the final version. Below please find responses to specific questions.
>
> **1. Claims regarding Verma et al. (2023):** Thank you for pointing this out. We agree with the reviewer that Verma et al. (2023) was the first to extend the surrogate losses proposed in (Verma & Nalisnick, 2022) and (Mozannar & Sontag, 2020) to the multiple-expert setting in L2D. We will definitely revise the claims accordingly throughout the paper.
>
> **2. Experiments:** Thank you for the suggestions. As recommended by Reviewer m1gt, we will aim to include experiments involving real-world experts, such as those considered in (Mozannar et al., 2023) and (Palomba et al., 2024), in the final version.
>
> Our paper primarily focuses on theoretically principled surrogate losses for learning to defer with multiple experts, grounded in realizable $H$-consistency and $H$-consistency bounds, within the learning theory framework. While we have empirically shown that minimizing our proposed loss functions outperforms the baselines in realizable settings and matches the best-known results in synthetic non-realizable settings—as predicted by our theoretical analysis—we recognize the importance of further empirical exploration. Accordingly, we plan to dedicate future work to conducting a more extensive empirical analysis in real-world settings, particularly in important scenarios where expert domains overlap (as suggested by Reviewer wQs7) and the data is heterogeneous (as suggested by Reviewer 798e).
>
> We will also include a more detailed discussion of the limitations of our empirical analysis and outline potential directions for future work in the final version. Nevertheless, we believe our theoretical contributions are significant and well-suited for acceptance in the theory track of the ICML conference.
>
> **3. Relation To Broader Scientific Literature:** Thank you for the suggestion. As recommended by the reviewer, we will incorporate and discuss literature on the application of deferral-based mechanisms in real-world machine-collaboration scenarios in the final version.
>
> **4. Introduction:** Thank you for the suggestion, we will seek to revise the introduction accordingly.
>
> **5. Figure 1:** The behavior for lower values of $q$ is similar to that observed for $q = 0.7$ and $q = 1$. Note that for any $q > 0$, Theorem 3.2 guarantees realizable $H$-consistency. We chose $q = 0.7$ and $q = 1$ because they correspond to standard choices in prior work—$q = 0.7$ is a commonly suggested value within the generalized cross-entropy family (e.g., Zhang & Sabuncu, 2018; Mao et al., 2024b), and $q = 1$ corresponds to the mean absolute error loss, as used in (Ghosh et al., 2017; Mao et al., 2024b).
>
> In this realizable distribution, we observe that our proposed surrogate losses achieve close to 100% system accuracy, while all other baselines fail to find a near-zero-error solution. This demonstrates that our surrogates are realizable $H$-consistent, whereas the compared baselines are not.
>
> **6. Table 1:** That's a good observation. We are not claiming that our proposed surrogate losses outperform baselines such as Verma et al. (2023) on non-realizable distributions. Rather, our results show that our surrogate losses match the performance of these state-of-the-art baselines in non-realizable settings, as both are supported by consistency bounds.
>
> The strength of our approach lies in the realizable setting, where our surrogate losses are guaranteed to be realizable $H$-consistent, unlike the baselines—as illustrated in Figure 1. Due to the specific realizable form of our surrogate losses, our model also defers significantly more and shows a greater tendency to select expert 2 compared to the baselines of Verma et al. (2023) and Mao et al. (2024b). This highlights a key effect of our proposed surrogate: it induces deferral behavior that is qualitatively different from prior approaches, particularly on the SVHN and Tiny ImageNet datasets.

---

> > ### Comment · Reviewer_wQs7 · 2025-04-08
> >
> > I would like to thank the authors for their response. I keep my positive score.

---

### Decision · Program_Chairs · 2025-05-01

**Decision:**

Accept (poster)

**Comment:**

This paper examines the problem of multi-expert learning to defer (single stage).  While previous work has derived consistent surrogate losses for the problem [Verma et al., 2023], they are not realizable H-consistent.  This paper addresses the issue, proving realizable H-consistency, H-consistency bounds, and Bayes-consistency.  All reviewers are in favor of acceptance, with the prominent criticisms being the reviewers found the contribution somewhat incremental and that the experiments could be made more real-world.  After reading the authors' rebuttal, I am convinced that these are not significant issues, especially as the paper's contribution is mostly about the learning theory.

As for my own small recommendation, the paper could do a better job of summarizing / organizing the contributions / limitations of previous multi-expert work (e.g. H-consistent or not, two-stage vs single stage, etc); maybe a table (with rows being related work, columns being L2D settings) could do this effectively.